# A scheduler for rhythmic gene expression

Dimos Gaidatzis [1,2,4✉], Maike Graf-Landua[1,3,4], Stephen P Methot [1,4], Michaela Wölk[1,3],
Giovanna Brancati [1,3], Yannick P Hauser[1,3], Milou W M Meeuse [1,3], Smita Nahar[1], Kathrin Braun [1],
Marit van der Does [1,3], Sirisha Aluri[1], Hubertus Kohler [1], Sebastien Smallwood[1] &
Helge Großhans [1,3✉]

## Abstract

Genetic oscillators drive precisely timed gene expression, crucial for development and physiology. Using the *C. elegans* molting clock as a model, we investigate how oscillators can schedule the orderly expression of thousands of genes. Single-cell RNA sequencing reveals a broad peak phase dispersion in individual tissues, mirrored by rhythmic changes in chromatin accessibility at thousands of regulatory elements identified by time-resolved ATAC-seq. We develop a linear model to predict chromatin dynamics based on the binding of >200 transcription factors. This identifies nine key regulators acting additively to determine the peak phase and amplitude of each regulatory element. Strikingly, these factors can also generate constitutive, non-rhythmic activity through destructive interference. Validating its power, the model accurately predicts the impact of GRH-1/Grainyhead perturbation on both chromatin and transcript dynamics. This work provides a conceptual framework for understanding how combinatorial, non-cooperative transcription factor binding schedules complex gene expression patterns in development and other dynamic biological processes.

**Keywords** Oscillator; Developmental Timing; *C. elegans*; Computational Modeling; Chromatin
**Subject Categories** Chromatin, Transcription & Genomics; Development

## Introduction

Life depends on dynamic, temporally coordinated gene expression. This is particularly evident for rhythmic phenomena that, like circadian rhythms, vertebrate somitogenesis, or nematode molting, involve hundreds or even thousands of genes, expressed in a defined order (Oates et al, 2012; Patke et al, 2020; Tsiairis and Großhans, 2021). At the core of these phenomena are genetic oscillators, or clocks, composed of a small set of core clock genes that are connected through feedback loops to generate periodic gene expression autonomously, without external rhythmic input. The core clock controls the expression of a larger set of "output" genes that execute specific functions at defined times.

Much attention has been given to understanding the mechanisms that achieve rhythmic activity with a particular periodicity. However, the execution of biological functions equally depends on phase control: the scheduling of expression of output genes such that for each gene, it peaks at a specific time in the cycle, its peak phase. In the conceptually simplest way, individual core clock genes can define the peak phase of individual output genes through direct regulation. Indeed, this mechanism explains the clustering in specific phases observed for circadian clock output genes, where three gene regulatory elements, recognized by core clock factors, dominate output gene regulation (Falvey et al, 1996; Gekakis et al, 1998; Harding and Lazar, 1993; Hogenesch et al, 1997; Preitner et al, 2002; Ueda et al, 2002; Ueda et al, 2005; Yoo et al, 2005). However, in *C. elegans* larvae, thousands of molting clock output genes are expressed in a highly choreographed manner, with robust phase-locking and across a continuum of peak phases (Hendriks et al, 2014; Meeuse et al, 2020). More core clock components could conceivably expand the accessible phase spectrum to generate this broad peak phase dispersion, and over 100 *C. elegans* transcription factors (TFs) are rhythmically expressed (Hendriks et al, 2014; Meeuse et al, 2020). Yet, this may alter period stability. Moreover, although the specific components of the *C. elegans* core clock and their wiring have not been well defined, available evidence does not support a function for most of the rhythmically transcribed TFs in the core clock (Hauser et al, 2022; Kinney et al, 2023; Meeuse et al, 2023; Stec et al, 2021; Wu et al, 2022).

Here, we use single-cell RNA-seq (scRNA-seq) to identify seven epithelial tissues in *C. elegans* larvae that each display a broad peak phase dispersion, supporting this as an inherent feature of the oscillator. Employing ATAC-seq, we uncover thousands of regulatory elements with rhythmically changing chromatin accessibility. Linear modeling reveals that DNA binding by a small set of nine TFs can explain a substantial fraction of these dynamics. Our modeling approach shows that broadly dispersed phases of rhythmic chromatin and gene expression can be achieved through additive TF activity, without a need for cooperative interactions. Counterintuitively, this mechanism can also generate non-rhythmic activity through destructive interference. We conclude that additive activity provides a parsimonious mechanism whereby a small set of

[1]Friedrich Miescher Institute for Biomedical Research (FMI), Basel, Switzerland. [2]SIB Swiss Institute of Bioinformatics, Basel, Switzerland. [3]University of Basel, Basel, Switzerland. [4]These authors contributed equally: Dimos Gaidatzis, Maike Graf-Landua, Stephen P Methot. ✉E-mail: dimosthenis.gaidatzis@fmi.ch; helge.grosshans@fmi.ch

TFs can act as a scheduler that achieves accurate phase control for rhythmic gene expression.

## Results

### Oscillating genes exhibit a broad peak phase dispersion in individual tissues

The broad peak-phase distribution of oscillating gene expression in *C. elegans* is striking but derives from the analysis of whole animal samples (Hendriks et al, 2014; Meeuse et al, 2020). As different combinations of tissue-specific expression patterns can generate the same bulk RNA-seq output (Fig. 1A), we wanted to evaluate oscillating gene expression at the tissue level. To explore gene expression dynamics in individual tissues, we performed single-cell RNA-sequencing (scRNA-seq) on populations of synchronized larvae collected and processed at different times of the L2 and L3 stage (18, 20, 22, and 24 h after plating at 25 °C). This approach allowed us to sample a full oscillatory period at a resolution of 2 h. We sequenced a total of ~90,000 cells at a median of 4,277 UMIs per cell and performed a 2D projection using uniform manifold approximation and projection (UMAP; "Methods") (Fig. 1B). After filtering and quality control, we obtained a total of 19 cell clusters to which we assigned tissue identity by correlation with gene expression profiles from a *C. elegans* tissue atlas (Cao et al, 2017) (Figs. 1B and EV1A; Appendix Fig. S1). We note that neurons were depleted, presumably due to the FACS-based cell isolation procedure.

Some of the clusters (3, 4, 8, 11) displayed a donut-like shape in the UMAP projection, hinting at repetitive gene expression dynamics (Fig. 1B). Since the nonlinear nature of UMAP could obscure some of the underlying circular structure, we chose to apply principal component analysis (PCA; Methods) to analyze the clusters for oscillations. Plotting PC2 over PC1 for each cluster revealed seven tissues with a circular structure based on visual inspection, suggesting that for these tissues, oscillations are the predominant dynamic expression feature (Fig. 1C). The seven tissues with dominant oscillatory gene expression (termed "oscillating tissues" in the following) were pharyngeal gland (cluster 3), seam cells (4), vulval precursor cells (8), non-seam hypodermis (11), excretory cell (12), pharyngeal muscle (22) and pharyngeal epithelia (30). These are all epithelial tissues, with pharyngeal muscle being a myoepithelial cell type. A nearly circular structure was also observed for the intestine, consistent with an enrichment for oscillating genes (Meeuse et al, 2020 and Fig. EV1B). However, this tissue also seems to have a more biased peak phase distribution, which might explain why the circle does not close. Thus, because tissues with insufficient cell numbers or more subtle oscillatory signals might not yield a circular structure in this analysis, we consider seven tissues as the lower bound, and a recent report identifies glial socket cells as another tissue with oscillatory gene expression (Weinreb et al., 2025).

We confirmed that the PCA projections represent temporal progression by labeling individual cells from the seam cell cluster according to their experimental timepoint (Fig. EV1C). This also revealed a large spread within timepoints. We speculate that cells partially continue their development during the lengthy dissociation process (2–3 h), leading to non-uniform distributions in time.

We concluded that pseudo-timing would be appropriate and thus inferred tissue-specific and time-resolved gene expression profiles for 36 timepoints, representing ~15 min intervals (see "Methods"). These expression profiles were synchronized using an external bulk RNA-seq reference (see "Methods" and Fig. EV1D–F), and biologically calibrated using expression profiles of molt entry and molt exit to align each dataset with the start of the larval stage, right after molt exit (see "Methods" and Appendix Fig. S2). This procedure resulted in a time-resolved expression atlas for the 7 oscillating tissues [these data are available as a resource for the *C. elegans* community—Dataset EV1).

Visualization of the gene expression data of the previously annotated oscillating genes, as a heatmap, provided three insights (Fig. 1D). First, expression peak phases were broadly dispersed even in individual tissues, a finding that we confirmed by visualizing the amplitude and peak phase for all genes per tissue, derived by cosine fitting on individual tissues (Fig. 1E). Second, most genes had a preferred tissue of expression, yet often exhibited oscillatory expression in additional tissues, at lower average levels but with broadly similar patterns. Finally, the profiles observed in the individual tissues appeared compatible with whole animal (bulk) sequencing.

We wanted to validate that our pseudo-timing approach recapitulated proper dynamics and that we had captured the major oscillating tissues. Hence, we combined the tissue-specific transcriptomes to reconstitute an in silico bulk time course dataset, weighing each tissue relative to the number of nuclei in larvae (Appendix Fig. S3A). In the case of oscillating tissues, pseudo-time expression data were used across the time course, while for the other tissues, the average expression of each gene was repeated at every timepoint. We then performed cosine fitting on the in silico reconstituted bulk dataset and compared the amplitudes to those previously obtained experimentally from bulk RNA-seq (Fig. 1F). The correlation between the two datasets was remarkably high ($R = 0.853$), confirming that the seven oscillating tissues produce most of the oscillating gene expression in developing worms. As a control, we compared each tissue alone against the bulk dataset, which yielded lower correlations (Appendix Fig. S3B). For a small gene set, the in silico reconstituted bulk did not recapitulate the experimentally observed oscillation (Fig. 1F, green dots). These genes were predominantly expressed in the intestine (Appendix Fig. S3C), which our analysis assumed to exhibit constant rather than oscillatory expression over time due to a lack of pseudo-timed expression data (see above). The genes that oscillate in both datasets (Fig. 1F, red dots) exhibited not only consistent amplitudes but also a remarkable correlation of phases (Fig. 1G)).

Taken together, our analysis shows that a broad peak phase distribution occurs even in individual tissues. Hence, it constitutes a genuine feature of the oscillator rather than a summation effect across tissues in whole animal data. Conversely, bulk analysis is suited to further investigation of oscillator mechanisms.

### ATAC-seq reveals oscillatory chromatin dynamics during larval development

We showed previously that oscillatory gene expression is generated through rhythmic transcription (Hendriks et al, 2014; Meeuse et al, 2023), yet the mechanism that can schedule thousands of genes to peak at such diverse times, in defined

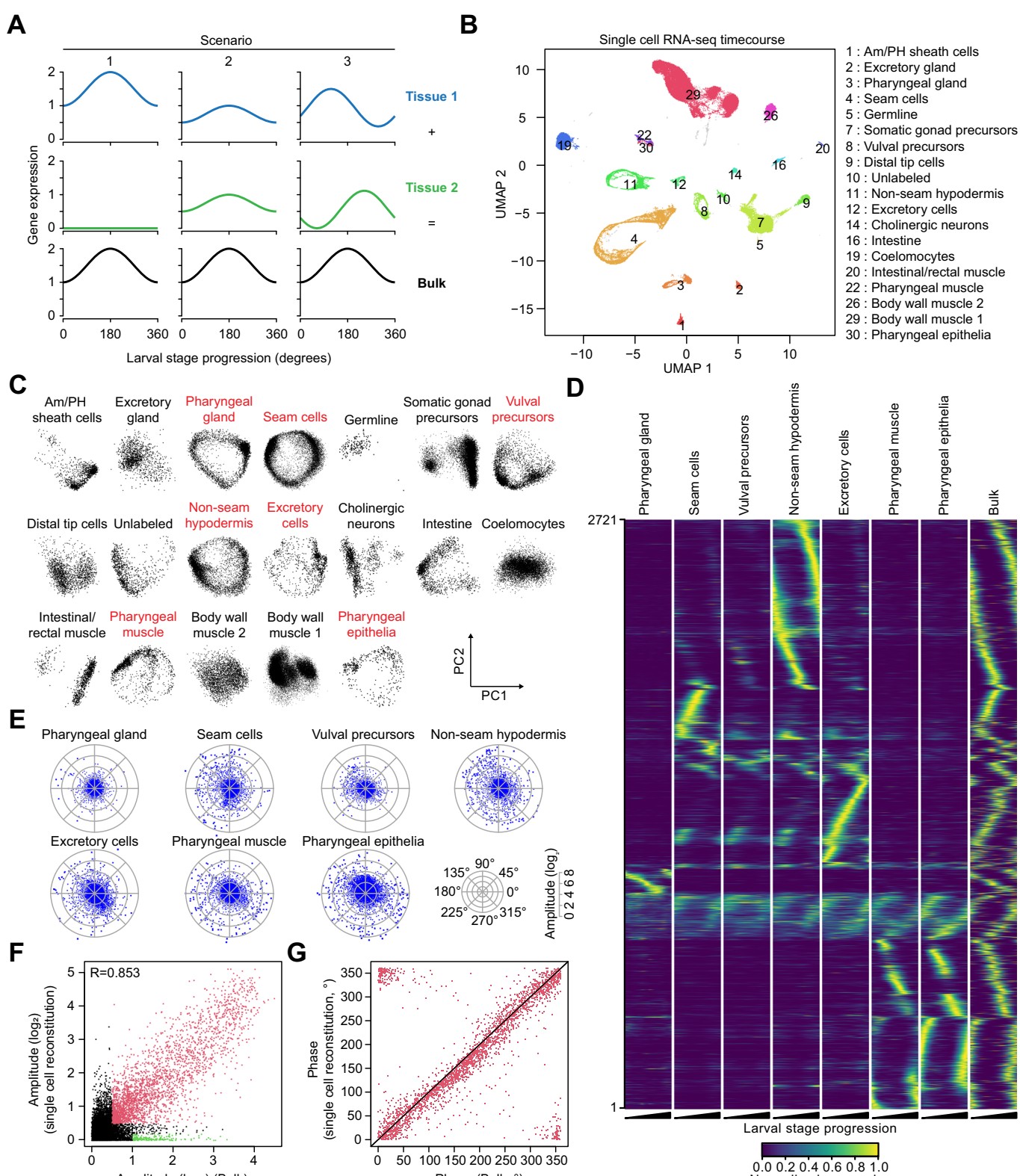

order, remains unclear. Hence, we sought to identify the regulatory elements that produce the observed transcription dynamics. Previous work showed that the exploration of chromatin accessibility using ATAC-seq (Assay for Transposase-Accessible Chromatin using sequencing) can identify developmentally relevant and tissue-specific regulatory elements in *C. elegans* (Daugherty et al, 2017; Durham et al, 2021; Jänes et al, 2018; Serizay et al, 2020). In two biological replicate experiments (performed independently), we sampled populations of synchronized larvae every hour over L2 and L3 stages (14–30 h

**Figure 1.    Individual *C. elegans* tissues display rhythmic gene expression with a broad peak phase dispersion.**

(A) Schematic drawing of wave superposition, showing how different combinations of tissue expression can generate the same bulk expression output for a given gene. For each column, gene expression output from Tissue 1 (blue) and Tissue 2 (green) are combined to generate the "bulk" expression pattern (black). (B) UMAP projection of scRNA-seq analysis with cells from all time points. Each dot represents a cell, positioned in the UMAP according to its expression profile. Numbered clusters passed quality control and were assigned to a tissue, except for cluster 10 which correlated with multiple tissues (see "Methods" and Fig. EV1A). (C) PCA scatterplots, individually comparing PC1 and PC2 for all cells from the respective indicated tissue. Tissues with circular expression dynamics are highlighted (red). (D) Heatmap representing max-normalized gene expression for all oscillating genes within oscillating tissues (rows), sorted by one-dimensional tSNE. Each column corresponds to a tissue, with expression values sorted along pseudo-time (beginning to end of a larval stage). The rightmost column visualizes the peak phases from the bulk RNA-seq data as squared normal distributions that peak at the previously determined phases for each gene (Meeuse et al, 2020). Color intensity indicates the relative expression level for each gene. (E) Tissue-specific gene expression radar charts plotting amplitude (radial axis, in log$_2$) and peak phase (circular axis, in degrees) as determined by cosine fitting for all genes in oscillating tissues. (F) Scatterplot comparing gene expression amplitudes (log$_2$) from the bulk RNA-seq dataset to the amplitudes from the in silico reconstituted scRNA-seq dataset. Red dots highlight genes oscillating in both datasets (log$_2$(Amplitude) > 0.5). Green dots highlight genes oscillating in the bulk but not the reconstituted dataset (see Appendix Fig. S3C). Pearson's correlation coefficient R is indicated. (G) Scatterplot comparing the peak-phases for genes that oscillate in both our bulk and in silico reconstitution datasets (red dots in F). Both datasets are molt-sync calibrated.

after plating at 25 °C). We then performed mRNA-seq and ATAC-seq on separate aliquots of each sample. Pairwise correlation plots revealed the previously described oscillatory pattern for the mRNA samples (Fig. EV2A). Strikingly, when analyzing 42,482 ATAC-seq peaks that we could identify, a similar pattern emerged, especially when focusing on highly variable (SD > 0.4) peaks (Fig. EV2B).

To define oscillating ATAC-seq peaks, we performed cosine wave fitting (in log$_2$ space). Pseudo-timing determined from the matched RNA-seq data (Fig. EV2C,D—see "Methods") allowed us to combine the two replicates in a single cosine fit and to calibrate the calculated phases such that zero degrees represents the start of a larval stage (Appendix Fig. S2). As in previous analyses for mRNA and miRNA, we included a graded component in the fit to observe peaks with a trend (i.e., monotonically increasing or decreasing signal over time) (Hendriks et al, 2014; Meeuse et al, 2020; Nahar et al, 2024). Plotting the amplitude over the graded component from the fit, and considering variability among replicate experiments, allowed us to use empirically determined cut-offs to classify ATAC-seq peaks into three distinct classes (Figs. 2A and EV2E–G; Dataset EV2). First, most peaks (31,436—74%) did not display any drastic changes throughout the time course and were categorized as *flat* (black in Fig. EV2F). A second subset (5853—14%) gradually changed in accessibility over time (*graded*, green). Finally, 5193 peaks (12%) with an amplitude over 0.25 were categorized as *oscillating* (red). Amplitudes were generally smaller for ATAC-seq peaks than for mRNAs but displayed a broad peak phase dispersion (Fig. 2B). Finally, we intersected the ATAC-seq peaks with a tissue-specific ATAC-seq atlas (Durham et al, 2021) and found that oscillating ATAC-seq peaks were generally enriched in the oscillating tissues identified by scRNA-seq (Appendix Fig. S4A).

ATAC-seq peaks were enriched around gene transcription start sites (TSS), allowing us to assign 25,461 peaks to genes by using a window of 2000 bp upstream of the TSS and 500 bp reaching into the gene body (Fig. EV2H). We then compared the top amplitude ATAC-seq peak per gene to the transcript level amplitudes, which revealed a correlation at the genome scale, as many data points fell along the diagonal (red points in Fig. 2C, n = 1413). Other points deviated from this relationship. The points along the y axis (n = 1195) likely reflect ATAC-seq peaks that were mis-assigned as the compact nature of the *C. elegans* genome entails that some peaks are assigned to more than one gene (Fig. EV2H). The points along the x axis (n = 1118) largely represent genes expressed in the pharynx (Appendix Fig. S4B), reflecting depletion of pharynx nuclei in the samples subjected to ATAC-seq (see "Methods").

When comparing the peak phases for all pairs of amplitude-matched ATAC-seq peaks and mRNAs, we could observe a high peak phase correlation (Fig. 2D). Notably, we detected a slight delay of ~23 min between chromatin dynamics and mRNA accumulation, which is similar to the 15 min delay between transcription and mRNA accumulation (Hendriks et al, 2014), suggesting that chromatin opening is accompanied almost immediately by transcription. Taken together, our results show that rhythmic mRNA production coincides with dynamic changes in chromatin accessibility for a large set of genes.

## ATAC-seq peaks identify functionally relevant sequences for oscillatory gene expression dynamics

To confirm that the oscillating ATAC-seq peaks could instruct dynamic gene expression, we focused on two virtually anti-phase peaks upstream and downstream of the *daf-6* locus (*upstream* - peak phase 124° [199 bp-long] and *downstream* — 335° [343 bp]) (Fig. 3A–C). The *downstream* peak is indeed close to the transcription start site of another gene, *nac-1*, whose transcript levels also peaks nearly in anti-phase to *daf-6* mRNA (*daf-6*—157°, *nac-1*—359°). We constructed reporter transgenes including either the full promoter sequence of each gene, or only the respective ATAC-seq peaks fused to a Δ*pes-10* minimal promoter. Each of these four promoter constructs was used to drive expression of a nuclear-localized, destabilized green fluorescent protein sequence (*pest::gfp::h2b*) (Fig. 3D). We quantified GFP levels of the corresponding single-copy transgenic worms using quantitative time-lapse imaging (Meeuse et al, 2020). All transgenes exhibited oscillatory expression. Moreover, and consistent with the distinct peak phases of the ATAC-seq peaks, expression oscillated in highly distinct phases for the two ATAC-seq peak reporter transgenes, while matching the phase and tissue specificity of their corresponding full-length promoter construct (Fig. 3E; Appendix Fig. S5A,B). Hence, these oscillating ATAC-seq peaks suffice to produce oscillatory gene expression in a specific phase, and they exhibit local activity.

## Binding of a subset of transcription factors is enriched on oscillating ATAC-seq peaks

To begin investigating which transcription factors could bind preferentially to these oscillating ATAC-seq peaks, we overlapped publicly available (Kudron et al, 2018, Data ref: Kudron et al, 2020) and internally generated ChIP-seq data from 284 transcription factors (out of 763 estimated in *C. elegans* (Narasimhan et al, 2015)) with the oscillating ATAC-seq peaks. Specifically, for each factor, we calculated

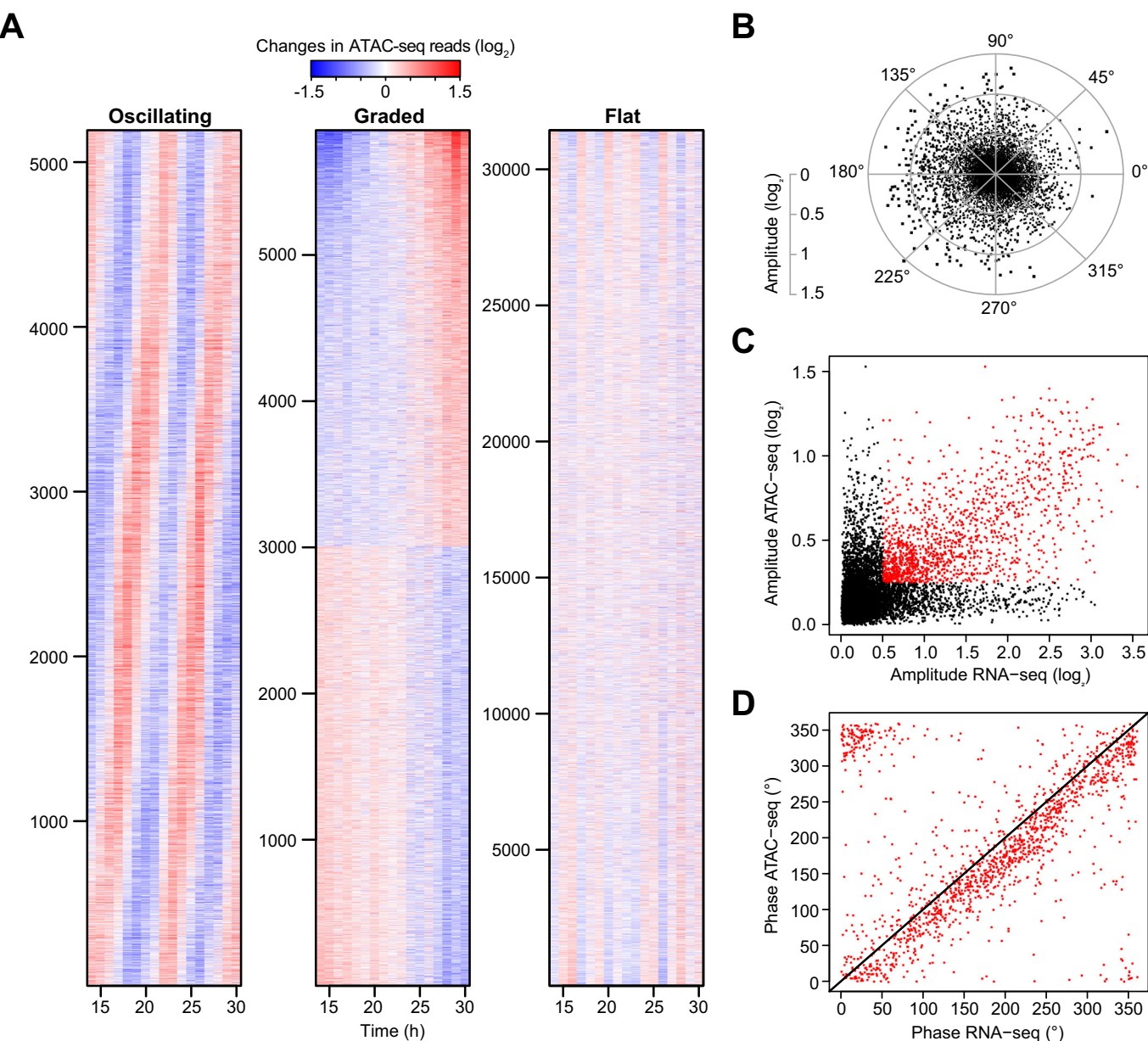

**Figure 2. Chromatin accessibility oscillates in phase with gene expression.**

(A) Heatmaps of mean-normalized and log₂-transformed ATAC-seq accessibility at individual peaks over time. Peaks were classified according to their amplitude and graded linear coefficient (Fig. EV2F). Oscillating peaks are sorted by their peak-phase, graded peaks by the graded component, and flat peaks by principal component 1. See Fig. EV2G for replicate experiment. (B) Chromatin accessibility radar chart plotting amplitude (radial axis) and peak phase (circular axis) as determined by cosine fitting for all ATAC-seq peaks. (C) Scatterplot comparing gene expression amplitudes (log₂) to accessibility amplitudes (log₂) for pairs of overlapping promoters and ATAC-seq peaks. In the case of multiple ATAC-seq peaks overlapping a given promoter, we selected the ATAC-seq peak with the highest amplitude. Note: each ATAC-seq peak can be assigned to multiple promoters due to genomic proximity (see Fig. EV2H). Red dots highlight pairings where both the ATAC-seq peak (log₂ > 0.25) and the gene expression (log₂ > 0.5) are considered oscillating. (D) Scatterplot comparing the peak-phases for oscillating pairs of ATAC-seq peaks and genes (red dots in C). The black line represents the diagonal.

the number of called ChIP-seq peaks that overlapped any of the ATAC-seq peaks, and then the fraction of those overlapping ATAC-seq peaks that were classified as oscillating. When displayed in a scatterplot, this revealed a subset of 29 TFs that were enriched for binding to oscillating ATAC-seq peaks above background (12.2%), while most TFs were strongly depleted (Appendix Fig. S6A). Notably, four out of six candidate core clock genes identified in a previous genetic screen, i.e., NHR-25, NHR-23, GRH-1 and BLMP-1, were among the most enriched TFs (Meeuse et al, 2023), as were NHR-85 and ELT-3, which have also been implicated in oscillatory gene expression (Kinney et al, 2023; Stec et al, 2021).

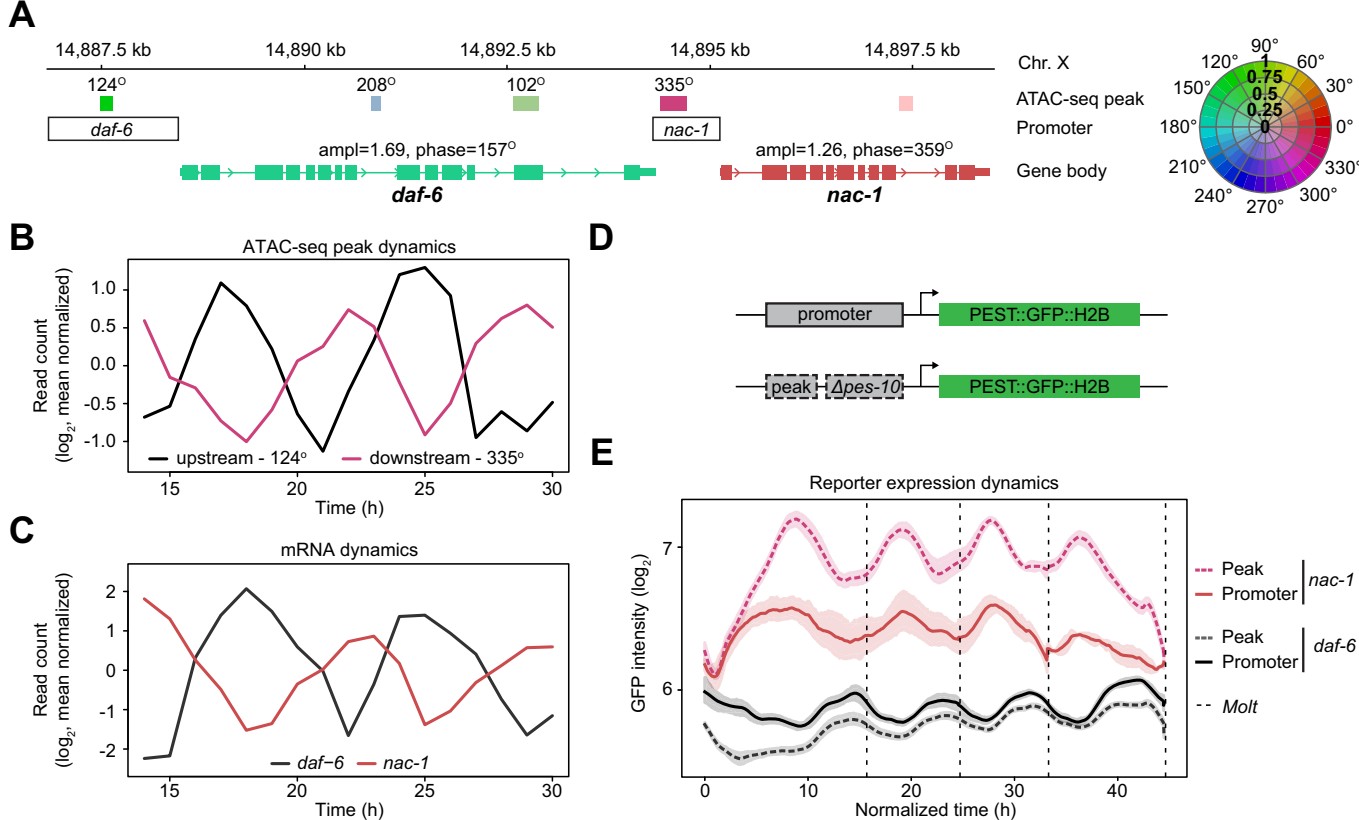

**Figure 3. Oscillating ATAC-seq peaks can act as phase-specific regulatory elements.**

(A) Schematic of the *daf-6*, *nac-1* locus on the X chromosome. All ATAC-seq peaks and gene bodies are shown as boxes with the color indicating peak-phase and the intensity indicating amplitude according to the color wheel on the right. For oscillating ATAC-seq peaks, the peak phase is indicated above. ATAC-seq peaks cloned for reporter assays are indicated above transparent black boxes representing the respective promoters. (B) Mean-normalized chromatin accessibility for the ATAC-seq peaks upstream and downstream of *daf-6* (black and red, respectively). (C) Mean-normalized gene expression for *daf-6* (black) and *nac-1* (red) mRNA. (D) Reporter lines were generated using either the full promoter or oscillating ATAC-seq peaks upstream of a minimal *Δpes-10* promoter as drivers of destabilized PEST::GFP::H2B. (E) Quantification of average GFP levels (lines) ± 95% confidence interval (shading) throughout larval development acquired by single worm imaging. Developmental time was normalized by aligning individual animals at each of the four molt exits (dashed lines). Number of animals analyzed from 1 replicate: Peak *daf-6* = 24, Promoter *daf-6* = 7, Peak *nac-1* = 15, Promoter *nac-1* = 7. Source data are available online for this figure.

## Linear modeling identifies potential core clock transcription factors

The preceding analysis cannot determine whether and how the 29 enriched TFs would contribute to chromatin oscillations. To address this issue, we tested whether TF-binding data is sufficient to predict the amplitude and phase of ATAC-seq peaks. To avoid confounding effects from non-oscillating tissues, we used a set of ATAC-seq peaks (hereafter referred to as Osc+) that only included oscillating peaks and an additional 753 non-oscillating peaks specific to oscillating tissues (Durham et al, 2021) (Fig. 4A). We devised two linear models, one that predicts the Cartesian coordinate *x* in the Fig. 4A, and a second that predicts *y*, using an approach similar to (Sobel et al, 2017). These linear models use a set of independent variables as predictors (transcription factor binding data) to calculate an output (*x* or *y*, calculated as a weighted sum of the independent variables) in order to maximize the correlation between the output of the model and the observed data (Fig. 4B). Note, we will use several iterations of these linear models and quantify their respective performance using the adjusted $R^2$, which considers the different number of parameters

used in each model. In our first model, we used the binding status (Occupancy matrix C, Fig. 4B) for a total of 213 TFs that overlap at least 10 peaks in our dataset, as our predictors. We found that this simple model could explain a substantial proportion of the total variance (adjusted $R^2$ of 0.305 and 0.335 in *x* and *y*, respectively). To identify the most influential TFs in this model, we combined their individual *P* values for x and y using Fisher's method. Ranking the TFs for significance revealed a heavily polarized distribution, with NHR-25, NHR-23, GRH-1 and BLMP-1, factors known to affect molting timing, being the highest-ranked TFs, and additional factors with no previously demonstrated roles in molting timing, LIN-14, NHR-43, HLH-11, NHR-21 and HAM-2 also standing out (Fig. 4C). These nine TFs were also enriched for binding to oscillating peaks and generally expressed in oscillating tissues (Appendix Fig. S6A,B).

From here on, we will only focus on these nine TFs and refer to them as the "molting clock TFs". Nonetheless, we wish to stress that, at this point, all nine factors should be considered as proxies of activity. This is because in the absence of a fully comprehensive collection of ChIP-seq data, we cannot exclude that the factors that

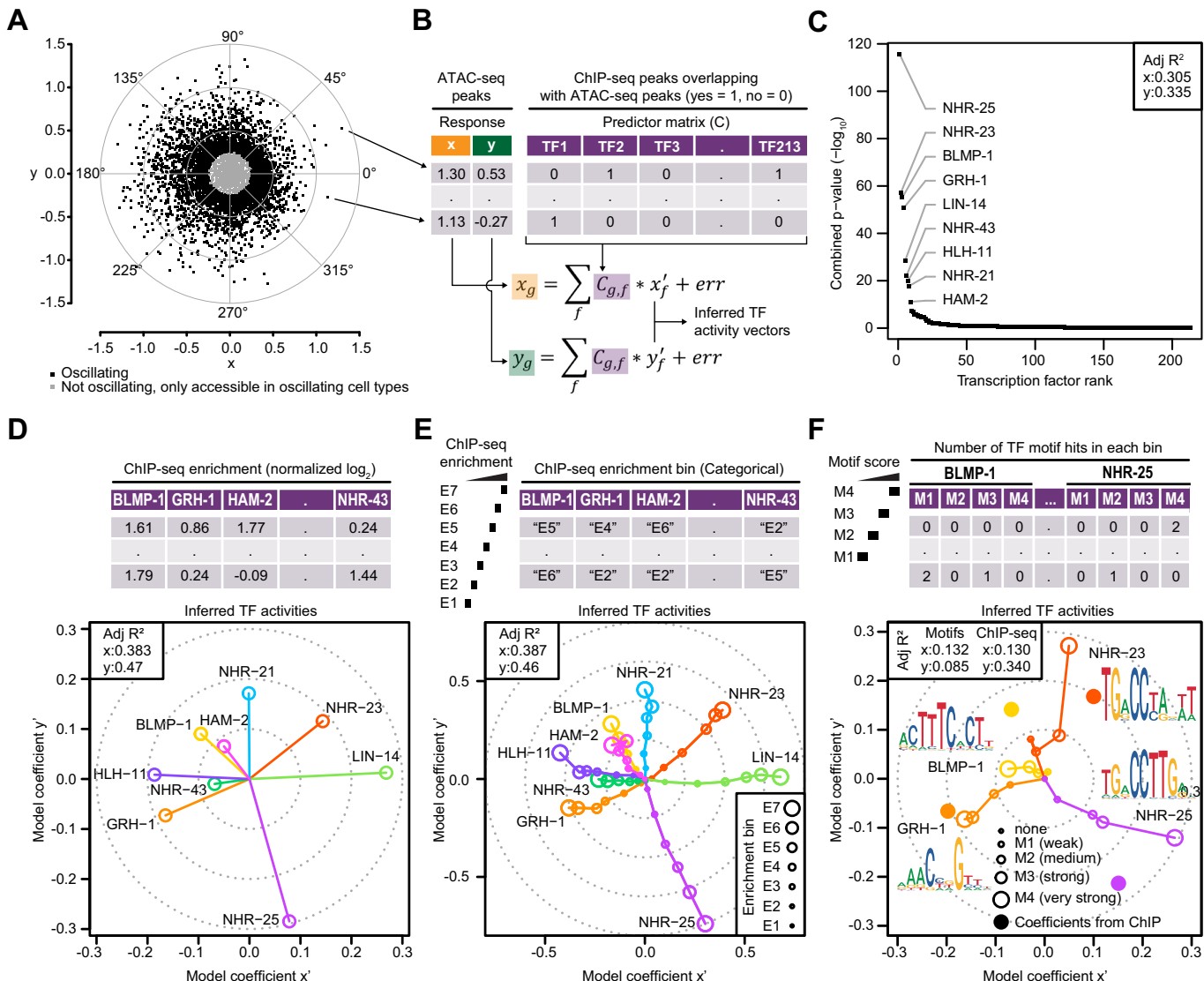

**Figure 4. Linear modeling identifies 9 transcription factors that can describe chromatin dynamics.**

(A) Radar plot (as in Fig. 2B) showing only oscillating ATAC-seq peaks and non-oscillating ATAC-seq peaks that are exclusively accessible in oscillating tissues (Durham et al, 2021). Plot contains both polar and Cartesian coordinates to highlight their interchangeability. (B) Scheme of the linear modeling approach that predicts x and y coordinates from (A) using binary TF occupancy data (from ChIP-seq experiments). Tables indicating the two responses (x and y coordinates) for each ATAC-seq peak, and the predictor matrix (occupancy of 213 TFs that indicates whether any ChIP-seq peak of a particular TF overlaps with a given ATAC-seq peak) are presented for visual clarity. The equations used for the two linear models are indicated below: C refers to the TF occupancy matrix, g refers to the genomic region, f refers to the TF, x' refers to the TF activity vector for the x component, and y' refers to the TF activity vector for the y component. After fitting the two models to the data, the output is given by two activity vectors (model coefficients x' and y'). (C) Scatterplot indicating the relative importance of each TF for the two linear models. TFs are ranked in descending order along the x axis according to their combined P value (from the two models), which is indicated on the y axis. Labeled TFs passed the manually set threshold of $\log_{10}(P) < -10$. The predictive power of the model (adjusted $R^2$) is indicated in the top right corner. (D) A linear model that predicts x and y from quantitative ChIP-seq enrichment data using only the nine most significant TFs from (C). The approach is as in (B) except that the occupancy-based predictor matrix is replaced by a matrix that contains $\log_2$ ChIP-seq enrichment values (ChIP vs input within a given ATAC-seq peak). The inferred TF activities are visualized as vectors (model coefficients x' and y'). The layout of the coordinate system is analogous to the one from (A). (E) A linear model that predicts x and y from binned ChIP-seq enrichment values. The approach is as in (D) but uses a predictor matrix that contains categorical variables instead of continuous variables. The categories are defined by stratifying the ChIP-seq enrichment values for each TF into seven equidistant bins. Each TF yields multiple activity vectors that are visualized as paths, starting at the origin and progressing in order of increasing enrichment. (F) A linear model that predicts x and y from TF motif (weight matrix) hits. For each of four molting clock TFs, the number of weak, medium, strong and very strong motif occurrences in ATAC-seq peaks were counted to yield a predictor matrix with $4 \times 4 = 16$ columns. Each TF yields multiple activity vectors that are visualized as paths, starting at the origin and progressing in order of increasing motif score (open circles). Coefficients from a linear model similar to (D) but considering only these four TFs (closed circles) are shown as a reference. (D–F) The predictive power of each model (adjusted $R^2$) is indicated in the top left corner.

we selected merely correlate with the binding of another, causally relevant factor that was not present in our data. This caveat is particularly relevant for factors for which we did not have L2/L3 stage ChIP-seq data (the time during which we performed ATAC-sequencing) and that, like LIN-14, are thought to be present only at low or even negligible levels during later larval stages (Ruvkun and Giusto, 1989; Wightman et al, 1993).

## Molting clock TF binding strength correlates with predicted activity

Given that the TF binding data used so far was only binary (binding/no binding), we wondered if we could improve the model performance by incorporating quantitative information in the form of normalized ChIP-seq enrichment values for the nine molting clock TFs (see "Methods"—Dataset EV3). Despite the greatly reduced number of TFs, relative to the occupancy model with 213 TFs, the new model showed substantial improvement in the total predictive power (adj. $R^2$ of 0.383 for $x$ and 0.470 for $y$). To visualize the contributions of each TF within one developmental cycle, we plotted their two coefficients as a vector, i.e., their relative contribution for predicting the $x$ or $y$ coordinates of each ATAC-seq peak. We found that the predicted activities of the molting clock TFs were well distributed around a circle (Fig. 4D). For example, LIN-14 showed peak activity at the start of a larval stage (~0°), while NHR-25 peaked at molt entry (~275°) (Fig. 4D).

To elucidate the relationship between the ChIP-seq binding strength of a TF and its contribution in the model, we created a new model after binning the ChIP-seq enrichment values for each TF into seven regularly spaced binding strength levels (Fig. 4E). For each TF, this revealed a trajectory that stretched out from the center (weakest binding) to the periphery (strongest binding) in a monotonous manner (Fig. 4E). Notably, this binning approach did not improve model performance (adj. $R^2$ of 0.387 for $x$ and 0.460 for $y$), suggesting that the activity of the TFs increases proportionally to their binding enrichment.

To investigate whether the continuous TF binding enrichment that we observe at oscillating ATAC-seq peaks is dictated by DNA sequence, we used the independently determined motifs for the known mammalian homologs of the molting clock TFs NHR-23 (RORA), BLMP-1 (PRDM1), GRH-1 (GRHL2) and NHR-25 (NR5A) (from the Homer database (Heinz et al, 2010)) (Appendix Fig. S6C). We found that their motif strengths correlated with ChIP-seq enrichment (Appendix Fig. S6D). To test whether this sequence information alone was predictive for ATAC-seq dynamics, we created a new model using bins based on TF motif strength. Similar to the ChIP-seq model, the relative position of all 4 TFs remained consistent, and individual TF trajectories stretched out from the center following their motif strength (Fig. 4F). Furthermore, despite only including four factors, the model could still explain a proportion of the total variance (adj. $R^2$ of 0.132 for $x$ and 0.085 for $y$). By comparison, a model based on ChIP-seq and only using these same 4 factors achieved adj. $R^2$ of 0.130 for $x$ and 0.340 for $y$. Consistent with the notion that the available ChIP-seq datasets might miss certain relevant clock TFs, we identified phase-biased motifs, only a subset of which could be assigned to molting clock TFs, through an unbiased 5-mer analysis (Appendix Fig. S6E). Taken together, these data indicate that the underlying DNA sequence at oscillating ATAC-seq peaks at least partially dictates

recruitment of the relevant TFs in the appropriate amounts to generate rhythmic accessibility.

## Additive activity of TFs can largely explain the observed peak phases

In our current iteration of the linear model, co-bound molting clock TFs are assumed to act independently of one another to generate chromatin accessibility—cooperative effects between transcription factors are not considered. Nonetheless, cooperative interactions among TFs are a recurrent feature in developmental processes (Reiter et al, 2017), so we wondered if we could improve the predictive power of the model by incorporating additional parameters that capture cooperative effects. To do so, we extended the model by including interaction terms between all possible pairs of molting clock TFs. This would test if the co-binding of two TFs to a particular region would lead to dynamics beyond their independent contributions. Surprisingly, this had little benefit (adjusted $R^2$ of 0.417 for $x$ and 0.496 for $y$, Fig. EV3A), suggesting that, to a first approximation, the molting clock TFs act in an independent manner. Hence, we focused our further analyses on the linear model without interaction terms.

We note that despite the broad peak phase dispersion of oscillatory ATAC-seq peaks, the linear model achieves substantial predictive power with only nine factors that themselves have a biased phase distribution, with only two factors (NHR-25, LIN-14) having peak activity between ~220° and ~45° (Fig. 4D). Hence, it seemed unlikely that a broad peak phase dispersion was the result of each factor individually being responsible for the oscillations observed in a specific phase window. We hypothesized that linear additions of the TF activity vectors could resolve this conundrum, consistent with a phase vector model previously postulated to explain the expression pattern of the circadian clock gene *Cry1* (Ukai-Tadenuma et al, 2011). Specifically, in the case of multiple co-bound TFs at a given locus, the output is calculated by the sum of all the activity vectors of the respective TFs (Fig. 5A). For such a scenario to generally explain the wide peak phase dispersion, the molting clock TFs would need to exhibit extensive co-binding. To evaluate this, we used k-means clustering to separate all oscillating ATAC-seq peaks into 100 clusters, based on the binding of the molting clock TFs (Dataset EV3). Visualizing these clusters revealed diverse combinations of molting clock TFs, with very few bound by only a single TF (Fig. 5B). Notably, ATAC-seq peaks within most clusters had a clear peak-phase bias (Fig. EV3B).

To test how well the model allowed us to predict phase and amplitudes of ATAC-seq peaks for each cluster, we re-fit the model using only 10% of the data (maintaining the amplitude distribution) and compared its predictions to the observations made with the remaining 90% of the data (Dataset EV3). The TF coefficients from the reduced set of data were very similar to those obtained with the full set (Fig. EV3C). Moreover, when comparing predictions and observations for each cluster, we observed a very high correlation of 0.90 ($x$) and 0.96 ($y$) between the model and the experimental data (Fig. EV3D).

The model allows us to retrieve the individual contributions of the molting clock TFs for every cluster (Dataset EV3). We can visualize this by plotting a vector path for a given cluster (Figs. 5C and EV4). In many cases, the predicted output vector and observed

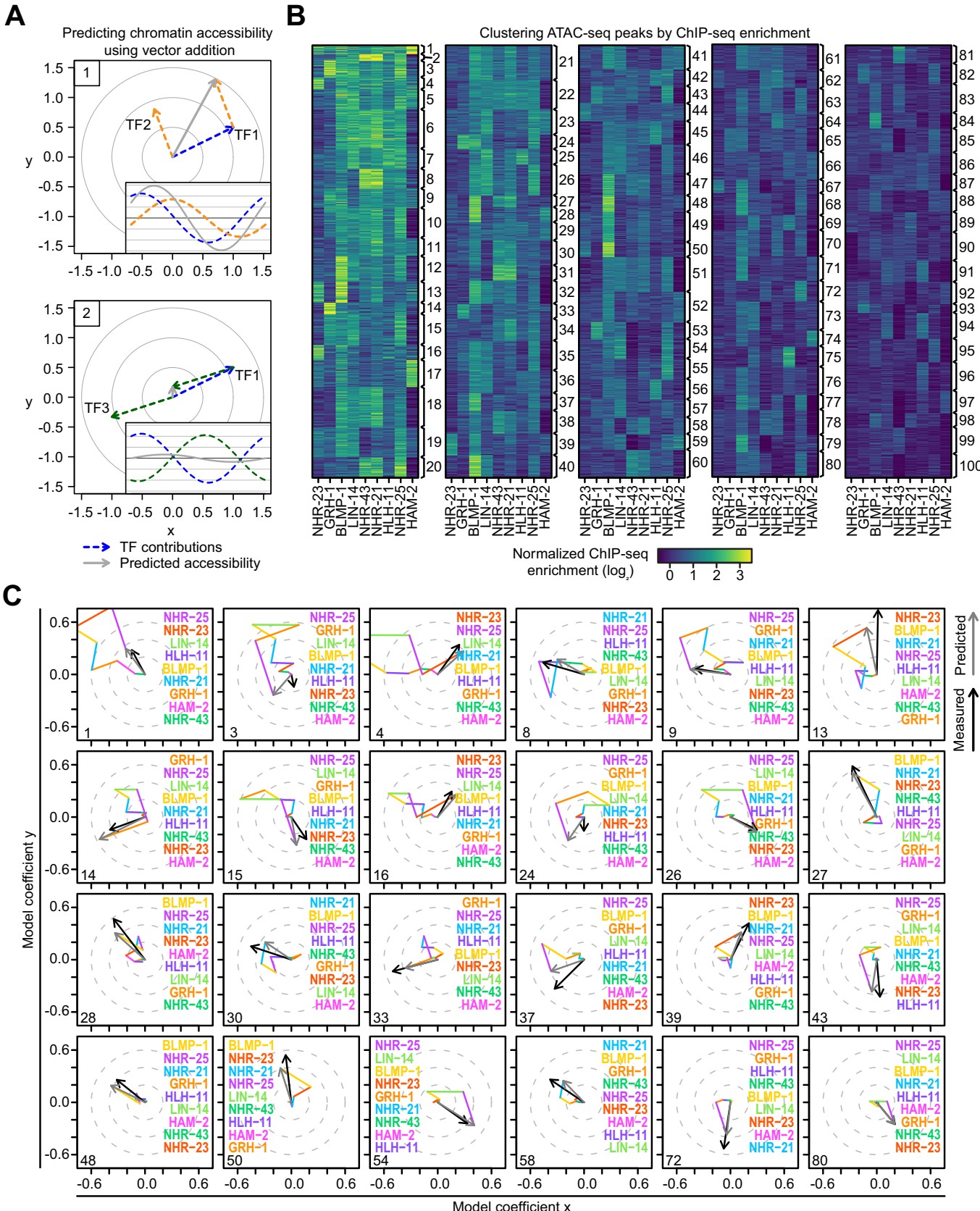

◄    **Figure 5.   Binding of 9 transcription factors is predictive of ATAC-seq phase and amplitude.**

(A) Schematic illustrating the additive effect of two TFs, each represented as an activity vector. Scheme 1 demonstrates how TF additivity can result in a higher amplitude output with an intermediate phase. Scheme 2 demonstrates destructive interference, whereby the summed activity of two TFs leads to a damped oscillation. Insets visualize the corresponding sinusoidal signals in time. (B) Heatmap showing occupancy of the nine molting clock TFs on ATAC-seq peaks (from Fig. 4A), organized into 100 clusters by k-means clustering according to the ChIP-seq enrichments. Clusters were organized according to the average enrichment of all 9 TFs (highest to lowest) and separated into five subgroups for visual clarity. (C) Phase-vector predictions (gray arrows) for individual ATAC-seq clusters were generated by summing up the phase-vector contributions (colored lines—average TF enrichment*x/y coefficients) of each TF. For each cluster, the phase vectors are added together in ascending order, starting with the shortest vector, while the list of TFs is organized in descending order. The average measured output vector for the cluster is indicated (black arrow). The top 24 clusters according to predicted output vector length are included here (see Fig. EV4 for the rest), ordered left to right, top to bottom.

peak phase are predominantly driven by the activities of two TFs, with the peak phase falling in between their respective activity vectors. Two clear examples are cluster 13, NHR-23 and BLMP-1, and cluster 54, LIN-14 and NHR-25. Yet, the predicted output vector of some clusters involves a complex combination of TFs (e.g., cluster 1). We conclude that the additive activity of different TFs is a major mechanism that explains the broad peak-phase dispersion that we observe. We note that the additive TF activity in the linear model operates in log space, in linear space this would correspond to multiplicative TF activity.

## Destructive interference between TFs can negate oscillations

Upon visual inspection of the vector plots (Figs. 5C and EV4), we found that many clusters follow a long and convoluted path before returning towards the center and arriving at their predicted output vector (e.g., clusters 1, 3, 4). To test if this observation holds true at the level of individual ATAC-seq peaks, we compared total path length (sum of all the lengths of the vector components) to the length of the predicted output vector (distance of the final prediction from the center) (Fig. EV3E). This revealed a nonlinear relationship between the two, with the predicted output vector length saturating at a total path length of ~1. This was not the case for a randomized control (see "Methods"). This suggests that beyond a certain threshold, increased binding of molting clock TFs does not necessarily drive higher amplitude oscillations. This is in line with the phenomenon of wave superposition (Fig. 5A, scenario 2), as anti-phase activities can cancel each other out by destructive interference. Indeed, many clusters have little to no oscillatory output despite substantial molting clock TF binding (e.g., clusters 2, 6, and 7 in Fig. EV4). Furthermore, we observed that the predicted output vector for several clusters (4, 14, 15, 28, 33, 46, 48, 55, 71, and 84) were predominantly driven by a single TF, despite additional TF binding, as the other TFs seemingly cancel each other out (Figs. 5B and EV4). Taken together, we find that destructive interference amongst molting clock TFs is a major constituent of the regulatory network.

## Acute GRH-1 depletion affects other molting clock components

To challenge our model rigorously, we wanted to test whether we could predict changes upon conditional depletion of one of the molting clock TFs, GRH-1. We had previously tagged endogenous GRH-1 with an auxin-inducible degron and found that auxin addition at the beginning of the L2 stage (21 h after plating at 20 °C) depletes GRH-1 rapidly (≤1 h) and causes an extension and aberrant execution of the M2 molt that terminates this stage, followed by death (Meeuse et al, 2023). We performed two independent biological replicates with matched RNA-seq and ATAC-seq at hourly resolution over 11 h (from 21 to 31 h after plating, at 20 °C) for control (vehicle—ethanol) and depletion (auxin) conditions. Replicates showed high correlation, allowing us to combine them (Appendix Fig. S7A). Consistent with earlier observations (Meeuse et al, 2023), timing analysis of the RNA-seq samples confirmed that depletion of GRH-1 caused premature termination of development before completion of a full oscillation cycle (Appendix Fig. S7B). Since our model assumes an intact developmental cycle (as is the case in wild-type animals), we cannot directly apply it to the GRH-1 depletion data. Therefore, we mean-normalized the data to one cycle of the control condition and applied the model to each timepoint independently, providing us with timepoint-specific coefficients (i.e., proxies of TF activity) for each of the molting clock TFs (Fig. 6A), similar to (Rey et al, 2011).

We compared the inferred TF activities from the control condition to those calculated upon GRH-1 depletion (Fig. 6B). Consistent with the effective depletion of GRH-1 protein, the model revealed greatly reduced GRH-1 activity, which occurred as early as 1 h after auxin addition. The model also revealed that GRH-1 depletion subsequently affected the activities of additional core clock TFs. For example, NHR-23 activity failed to decline in the manner observed in control animals, which occurred early on, approximately a quarter of the way through the larval stage. Noticeable effects also occurred for NHR-25 (impaired increase in activity), NHR-21, LIN-14, and HAM-2 (impaired decrease). Hence, at later time points, the effects of GRH-1 depletion will be a combination of direct and indirect effects. We conclude that larval arrest and death upon GRH-1 depletion are accompanied by widespread effects on the molting clock TFs, likely reflecting a collapse of the underlying oscillator.

## The model predicts the impact of acute GRH-1 depletion on chromatin accessibility

Next, we asked whether the time-dependent model could predict specific changes in chromatin accessibility upon GRH-1 depletion. Hence, we used 10% of the data to infer TF activities and then applied the model to the remaining 90% to predict changes for the different time points (Appendix Fig. S7C). At early time points, when there are few observable changes, correlations between predicted and measured

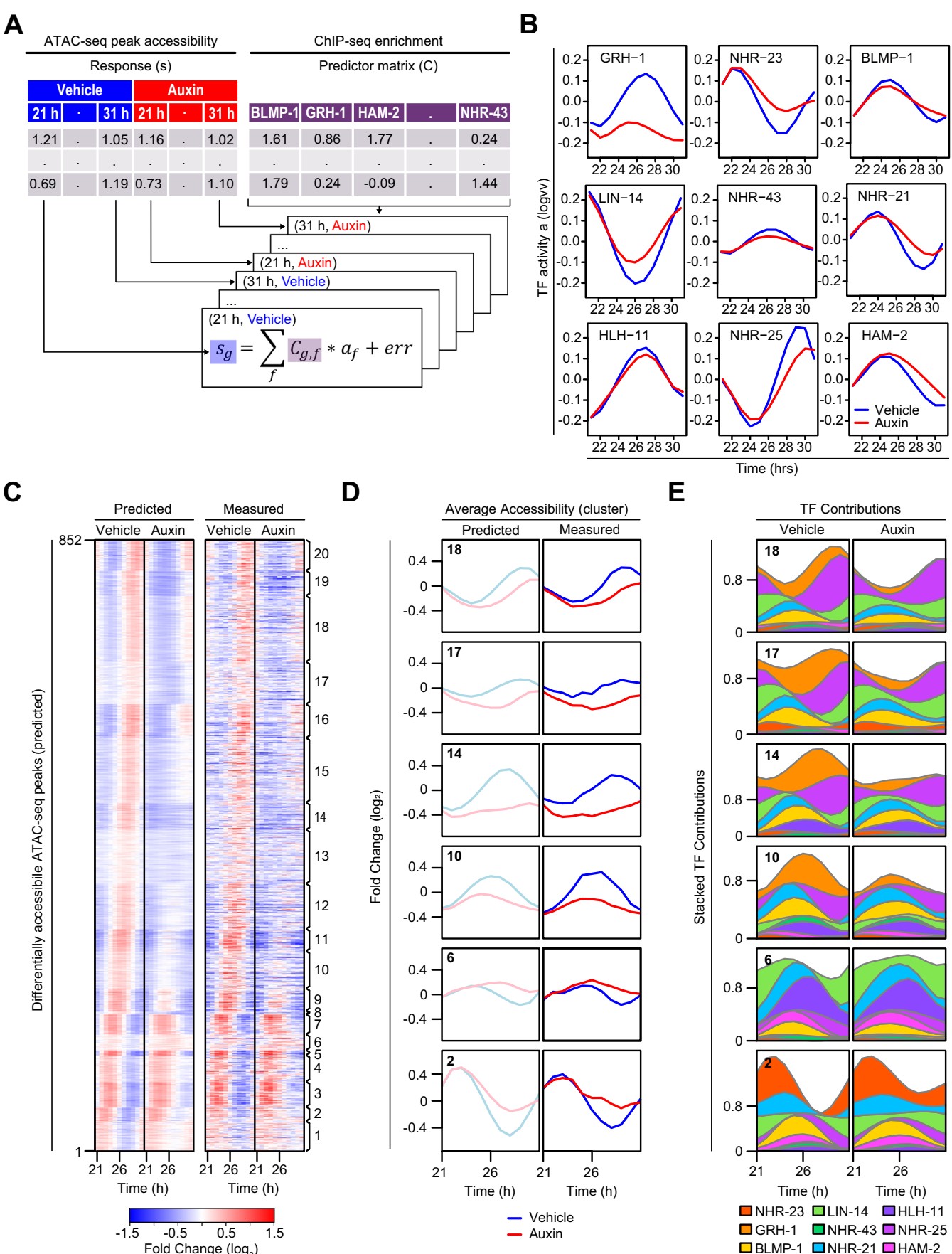

**Figure 6. A linear model can predict chromatin accessibility changes upon acute GRH-1 depletion.**

(A) Scheme of the linear modeling approach that predicts chromatin accessibility (peaks from Fig. 4A) at each timepoint using TF enrichment data (from ChIP-seq), applied to the ATAC-seq time course performed using *grh-1::degron* transgenic animals treated with either vehicle (ethanol) or auxin. Tables indicating the responses (accessibility of each ATAC-seq peak in each sample) and the predictor matrix (same as Fig. 4D) are included for visual clarity. The equation used for the linear model (one per timepoint and per condition) are indicated below: s refers to the sample, C refers to the TF enrichment matrix, g refers to the genomic region, f refers to the TF, a refers to the inferred TF activity. (B) Results from the linear modeling as described in (A). The *y* axis depicts the inferred TF activity for a given timepoint (vehicle—blue line or auxin—red line). Activity curves were smoothed using a spline regression with six components. (C) Heatmaps of mean-normalized and $\log_2$-transformed predicted (left) or measured (right) changes in ATAC-seq reads at individual peaks over time, for 666 peaks predicted to be differentially accessible. Differential accessibility was scored as $\log_2$(max absolute difference) >0.25 (See Appendix Fig. S7E). Peaks were organized into 20 clusters using k-means clustering, and then clusters were ordered, from bottom to top, by the average measured peak phase in N2, going from start to end of the larval stage. (D) Average predicted (left) or measured (right) accessibility changes over time for selected clusters of differentially accessible peaks, comparing vehicle and auxin treatments. (E) Plots compiling individual TF contributions, which generate the predicted chromatin accessibility over time. TFs are sorted by maximal activity difference between the vehicle and auxin conditions. (D, E) See Fig. EV5 to compare all clusters.

changes across ATAC-seq peaks were low (Appendix Fig. S7D). Yet, as the effect of GRH-1 depletion increases over time to yield more and larger changes, correlations also increased considerably up to a high $r = 0.605$ (Appendix Fig. S7D).

We focused on peaks with the largest predicted changes (Appendix Fig. S7E), and plotted heatmaps for the two conditions, vehicle and auxin, and for both predicted and measured changes (Fig. 6C). Clustering the heatmap revealed that the majority of ATAC-peaks (clusters 8–20) showed reduced accessibility in the auxin condition, while a smaller subset (clusters 1–7) showed increased accessibility. To compare the detailed changes in dynamics upon GRH-1 depletion, we aggregated the clusters to obtain meta-profiles (Figs. 6D and EV5A). For all clusters, the prediction matched closely to the measured data.

To gain insight into the mechanism underlying these changes, we plotted for each cluster the predicted activity patterns of the nine TFs in vehicle and auxin conditions (Figs. 6E and EV5B). This allowed us to visualize how the different combinations of molting clock TFs can drive dynamics over time. As expected, decreased GRH-1 activity was a major contributor for all clusters where opening was reduced upon auxin addition, i.e., clusters 8–20 (Figs. 6E and EV5B). For all other clusters, the altered pattern did not reflect a direct contribution of GRH-1. For instance, clusters 1 and 6 were dominated by altered activity of LIN-14, clusters 2–5 by NHR-23 and cluster 7 by HAM-2. In other words, the model can predict which mechanism(s) underlie altered ATAC-seq oscillations for the different clusters.

## The linear model can predict mRNA-level changes

Finally, we asked whether the model that predicts chromatin dynamics can also predict dynamics in gene expression. Using 1,117 high-confidence, phase-matched pairs of ATAC-seq peaks and genes from the original wild-type time course (Fig. 2D), we fit a new model replacing accessibility with the mRNA levels for these linked genes (Fig. 7A). The resulting TF activity profiles were highly similar to those obtained with ATAC-seq peaks (compare with Fig. 6B). We then compared the predicted gene expression changes to the observed changes and again as for the ATAC-seq peaks, found that correlations increased over time as the effects of GRH-1 depletion manifested (Fig. 7B,C). The predictive power using mRNA was only slightly lower than for chromatin accessibility. We conclude that the molting clock TFs function additively to direct the phase of both chromatin accessibility and gene expression, thereby controlling dynamic mRNA accumulation during larval development.

## Discussion

With thousands of genes exhibiting large amplitudes, oscillatory gene expression in developing *C. elegans* larvae is a striking phenomenon, which we have shown here to involve at least seven epithelial tissues. Each tissue displays a broad peak phase dispersion for the oscillating genes that it expresses, necessitating a robust scheduler to determine the specific time and order of expression for each output gene. Although ~100 TFs are rhythmically transcribed (Meeuse et al, 2020), we find that nine TFs, through additive activity, suffice to generate the full phase spectrum of output genes, allowing us to predict a substantial amount of the observed variance. Additional components likely exist, and cooperative TF activity may contribute in specific contexts (Kinney et al, 2023), yet our findings identify additive activity as a parsimonious mechanism to attain rhythmic gene expression with a broad peak phase dispersion through a small set of TFs.

We can speculate about two consequences of this scheduler design: first, as TFs act predominantly independently, the creation or destruction of a given binding site offers a simple mechanism to fine-tune phase and amplitude, facilitating evolutionary adaptation. Second, the ability to generate non-rhythmic outputs from rhythmic inputs, through destructive interference, could allow the same set of TFs to achieve both temporal dynamics and spatial specificity.

We propose that a scheduler based on linear additivity can also function in other rhythmic gene expression phenomena, especially circadian rhythms. Although circadian clock genes were previously reported to compete for regulation on circadian output genes rather than regulating them combinatorially (Fang et al, 2014), the additive activity of two distinct gene regulatory elements explains the unusual peak phase of the circadian core clock gene *Cry1* (Ueda et al, 2005; Ukai-Tadenuma et al, 2011). A mechanism of partial destructive interference may help to resolve the apparent contradictions: if multiple TFs of distinct, yet additive, peak phase activities bind to a given regulatory element, their effects on oscillations may cancel out to the extent that the resulting phase and amplitude mostly mirror the activity of a single, seemingly dominant, TF. Indeed, as suggested by mathematical modeling and synthetic biology experiments (Korenčič et al, 2014; Ukai-Tadenuma et al, 2008), an additive scheduler could also account for the differences in circadian peak phase distributions seen across animal tissues.

Finally, our study was directed at characterizing the scheduler that controls the output of the *C. elegans* developmental clock,

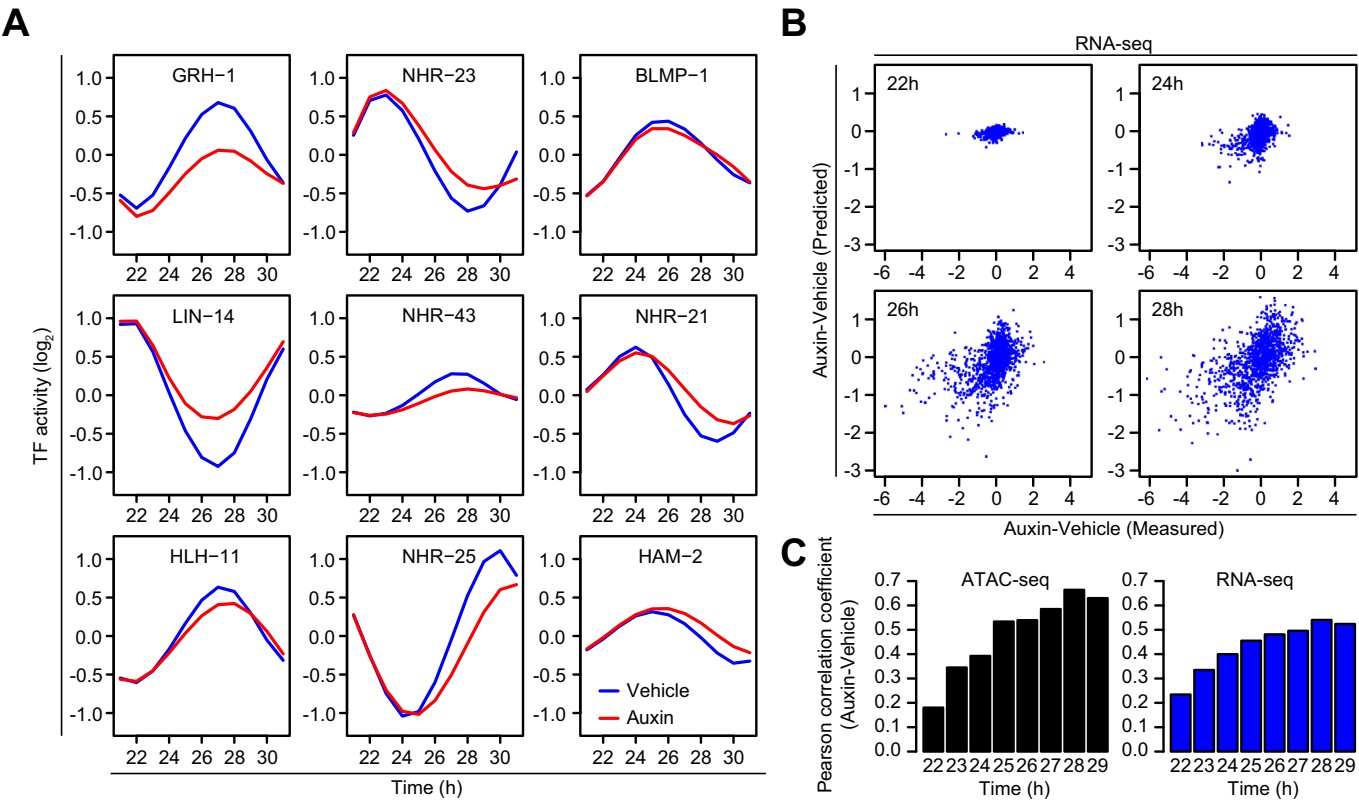

**Figure 7. Gene expression changes can be predicted using a chromatin accessibility model.**

(A) TF activities from the timepoint-based linear model (Fig. 6A), using RNA-seq measurements at genes with matching ATAC-seq peaks (see Fig. 2C—red dots). Time course performed using *grh-1::degron* transgenic animals treated with either vehicle (blue line) or auxin (red line). TF activity represents the model coefficient for each TF at each timepoint. (B) Scatterplots of differential RNA-seq (auxin—vehicle) at all oscillating genes with matched oscillating ATAC-seq peak, comparing predicted and measured changes to gene expression at indicated timepoints. Predictions were made using the ChIP-seq enrichments at linked ATAC-seq peaks and coefficients from (A). (C) Pearson correlation coefficients of predicted vs measured ATAC-seq or RNA-seq changes for each timepoint.

which may or may not be integral to the core clock itself. Yet, we note a striking overlap with genes previously identified as candidate core clock TFs, notably GRH-1, NHR-23, NHR-25, and BLMP-1 (Hauser et al, 2022; Meeuse et al, 2023; Stec et al, 2021). The evidence for GRH-1 as a core clock gene appears particularly strong, as we find that its depletion affects, directly or indirectly, the activity of other candidate core clock genes such as NHR-23. Hence, our observations provide a first glimpse into the potential wiring of the core oscillator and may guide future mechanistic studies on the molecular interactions between clock components.

# Methods

### Reagents and tools table

| Reagent/resource | Reference or source | Identifier or catalog number |
|---|---|---|
| **Experimental models** | | |
| *xeSi296[eft-3p::luc::gfp::unc-54 3′UTR, unc-119( + )] II* | Meeuse et al, 2020 | HW1939 |

| Reagent/resource | Reference or source | Identifier or catalog number |
|---|---|---|
| *oxSi221[eft-3p::GFP] II; xeSi449[eft-3p::mCherry::luciferase unc-119( + )] III* | this work, cross of *oxSi221* from CGC and *xeSi449* from (Stojanovski et al, 2022) | HW3125 |
| *xeSi159[daf-6p::pest::gfp::h2b::unc-54 3′UTR; unc-119 + ] II; xeSi449[eft-3p::mCherry::luciferase unc-119( + )] III* | This work | HW3507 |
| *xeSi165[daf-6(ATACpeak)::Δpes-10p::pest::gfp::h2b::unc-54 3′UTR; unc-119 + ] II; xeSi449[eft-3p::mCherry::luciferase unc-119( + )] III* | This work | HW3508 |
| *xeSi587[nac-1p::pest-gfp-h2b::unc-54 3′UTR, unc-119( + )] II; xeSi449[eft-3p::mCherry::luciferase unc-119( + )] III* | This work | HW3718 |

| Reagent/resource | Reference or source | Identifier or catalog number |
|---|---|---|
| xeSi588 [nac-1(ATACpeak)::Δpes-10p::pest::gfp::h2b::unc-54 3'UTR, unc-119( + )] II; xeSi449[eft-3p::mCherry::luciferase unc-119( + )] III | This work | HW3719 |
| nhr-23(wrd8[nhr-23::GFP::degron::3xFLAG]) | Ragle et al, 2020 | JDW29 |
| grh-1(xe135[grh-1::degron::3xFLAG]) I; xeSi296 [eft-3p::luciferase::gfp::unc-54 3'UTR, unc-119( + )] II; xeSi376[eft-3p::TIR1::mRuby::unc-54 3'UTR, unc-119( + )] III | Meeuse et al, 2020 | HW2434 |
| **Recombinant DNA** | | |
| **Antibodies** | | |
| Rabbit polyclonal anti-GFP | Abcam | ab290 |
| **Oligonucleotides and other sequence-based reagents** | | |
| Primers | This work | Appendix Table S1 |
| **Chemicals, enzymes, and other reagents** | | |
| cOmplete™, EDTA-free Protease Inhibitor Cocktail | Roche | 11873580001 |
| Proteinase (Pronase E) | Sigma-Aldrich | P8811-100mg |
| Firefly D-Luciferin | PJK biotech | 102111 |
| Digitonin | Sigma-Aldrich | D141-100MG |
| Tween-20 | Sigma-Aldrich | 11332465001 |
| Tn5 Transposase | In house | NA |
| NEBNext High-Fidelity PCR Master Mix | NEB | M0541S |
| AMPure XP beads | Beckman Coulter | A63881 |
| TRI Reagent® | MRC Inc | TR 118 |
| μ-slide 2 Well ibiTreat | Ibidi | 80286 |
| protein G Dynabeads | ThermoFisher Scientific | 10003D |
| 384-well plate, white | Berthold | 32505 |
| **Software** | | |
| ImageJ | https://imagej.net/ij/ | 1.54 |
| Excel | Microsoft | 16.97 |
| Illustrator | Adobe | 29.5.1 |
| **Other** | | |
| Wheaton stainless-steel Dounce Dura-Grind Tissue grinder | DWK Life Sciences | 357572 |
| FastPrep-24™ 5 G bead beating grinder and lysis system | MP Biomedicals | 6005500 |
| Bioruptor® Plus sonication | Diagnode | B01020001 |
| Luminometer Centro XS³ LB960 | Berthold | 46970-50 |
| Direct-zol RNA microprep kit | Zymo Research | R2062 |
| minElute PCR purification kit | Qiagen | 28004 |
| ChIP DNA Clean and Concentrator Kit | Zymo Research | D5205 |

### *Caenorhabditis elegans* strains

The Bristol N2 strain was used as the wild-type. Transgenic lines are listed in the Reagents and Tools table.

Promoter reporter lines were generated by taking 2000 bp of the upstream region of the respective gene, starting at the ATG. Alternatively, the upstream region up to the next gene was selected. Gibson assembly (Gibson et al, 2009) was used for cloning of the promoter region amplicon into the target vector (pYPH0.14 generated by *Not*I digestion of pCGJ150 and addition of *Nhe*I-digested *PEST::gfp::h2b)*. For ATAC-seq peak reporter lines, the sequence of the identified peak was used as a promoter and cloned into pYPH51 (pYPH0.14 backbone with Δpes-10::pest::gfp::h2b). Subsequently, the vector was injected into EG8079 worms containing the universal ttTi5606 locus on chromosome II and integrated through Mos1-mediated single copy insertion (MosSCI) (Frøkjær-Jensen et al, 2012; Frøkjær-Jensen et al, 2008). Worms were backcrossed to N2 worms at least three times.

#### Gibson assembly protocol

Primers with a ~20 nt overhang were designed to generate amplicons with 20 nt overhangs on both sides, which are identical to the left and right arm of the insertion site in the vector. 4 μL of the vector and 1 μL of the amplicon were incubated for 1 h at 50 °C with 15 μL of 1.33× Gibson master mix. Afterward, 4 μL of the mix were introduced in a DH5alpha *E. coli* strain and cultured. For injections, plasmids were purified with Midi-Prep.

## Animal growth conditions

*C. elegans* strains were maintained at 20 °C on Nematode Growth Media (NGM) plates seeded with OP50 bacteria and following standard protocols (Stiernagle, 2006).

To obtain large amounts of synchronized L1 stage larvae, ~30,000 worms were grown until the gravid adult stage on 15-cm Peptone-rich plates (25 g agar, 20 g peptone, 1.2 g NaCl, 975 mL $H_2O$, 25 ml PPB, 1 mL $MgSO_4$, 1 mL cholesterol (5 mg/mL in ethanol)), seeded with 10× concentrated OP50. Adults were bleached to isolate embryos, which were then allowed to hatch in M9 buffer, without food, and incubated overnight (≤18 h) at room temperature.

## Single-cell RNA-seq

*C. elegans* dissociation into single cells was performed according to the protocol from (Cao et al, 2017) with minor adjustments: 3,200,000 synchronized L1 HW3125 (*eft-3p::GFP; eft-3p::mCherry::luciferase*) worms were distributed on 15 cm peptone-rich plates at 200,000 larvae per plate. Worms were grown for 18, 20, 22, 24 h at 25 °C to the L2–L3 stages and then collected by thoroughly washing with M9 solution (3 plates per timepoint). Worms were washed until the supernatant was clear (≥3 times) and separated into multiple 1.5 mL low-bind tubes to ensure an ~100 μL worm pellet in each tube after centrifugation for 1 min at $16,000 \times g$. The worm pellet was resuspended in 200 μL of SDS-DTT solution (20 mM HEPES pH 8.0, 0.25% SDS, 200 mM DTT, 3% sucrose) to pre-sensitize the worm cuticle and incubated for exactly 4 min at RT. The reaction was then stopped by resuspension in 800 μL of egg buffer (25 mM HEPES pH 7.3, 118 mM NaCl, 48 mM KCl, 2 mM $CaCl_2$, 2 mM $MgCl_2$ with ~340 mOsm osmolarity). Worms were pelleted at $16,000 \times g$ for 1 min and resuspended in 1 mL of egg buffer. Washing steps were repeated four more times. After the final wash step, pellets were resuspended in 100 μL of freshly

thawed 15 mg/mL pronase E, diluted in egg buffer. Using a thinned-out glass Pasteur pipette, worms were dissociated by pipetting up and down until the worms were visibly dissociated (max 25 min). Dissociation was evaluated every 3 min until most worms were disrupted when viewed under a dissection scope. The reaction was stopped by adding 900 µL of L-15/10% FBS, gently mixed, and centrifuged for 5 min at $9600 \times g$ at 4 °C to pellet cells. The supernatant was carefully removed, and the wash steps were repeated two more times. After the last wash, the cells were resuspended in 1 mL of L-15/10% FBS and left to sit on ice for 30 min.

The supernatant (containing the cells) was carefully transferred onto a 30-µM filter and collected into sorting tubes. 2 µL of Draq7 dye per 1 mL of suspension was added to exclude dead cells. Using a BD FACSAria cell sorter, GFP and mCherry double-positive cells were sorted into 1.5-mL low-bind tubes. Cells were then diluted, assuming a typical 40% recovery, to generate libraries using the 10x Genomics protocol with the Chromium Controller (10×3′RNAv3). Subsequently, reads were sequenced using NextSeq HIGH-OUT 75 cycle paired-end sequencing.

## Pre-processing of scRNAseq data

scRNA-seq data was mapped to the *C. elegans* genome (ce11) and quantified (WormBase WS270 gene annotation) using CellRanger version 6.1.2 with default parameters. From a total of 107,700 cells, we retained 96,600 with a coverage of at least 750 UMIs. We further only considered 14,344 genes that were expressed in at least 0.05% of cells. We scaled the expression profile of every cell to the average library size, $\log_2$-transformed after adding a pseudocount of 1. We selected the top 2000 variable genes using the getTopHVGs() function from the scran package. For 2D embedding, we performed PCA (2000 top variable genes) using 30 components and calculated a UMAP using the function runUMAP() from the scater package. To obtain single-cell clusters, we segmented the UMAP as follows: We calculated a 2D density estimate using the function bkde2d() from the KernSmooth package (bandwidth = 0.15 on a $800 \times 800$ grid) and segmented the resulting 2D density into disconnected territories using the function bwlabel() from the EBImage package. After removing all clusters with less than 100 cells, we obtained a total of 22 clusters. Within this set we detected and removed four doublet clusters (Appendix Fig. S1— clusters 18, 23, 25, and 31), resulting in a total of 18 clusters.

## UMAP cluster annotations

To identify the cell types corresponding to the clusters, we compared pseudobulk expression profiles of the clusters to annotated pseudobulk expression profiles from public data (Cao et al, 2017) (Fig. EV1A). For most clusters, we identified a unique corresponding cell type, except for cluster 10, which we could not map, and cluster 22, which mapped well to both pharyngeal muscle and pharyngeal epithelia. Using gene markers from the published pseudobulk expression data (Cao et al, 2017), we managed to split cluster 22 into two components (Pharyngeal muscle: Cluster 22 and Pharyngeal epithelia: Cluster 30), resulting in a total of 19 annotated clusters.

## Detection of cell clusters with circular expression

To identify cell types with potential rhythmic gene regulation, we performed PCA on individual single-cell clusters as follows: For each cluster, we first normalized each cell to the average library size (considering only the cells of a particular cluster). Then we removed outlier cells with too many detected genes (1.5-fold over median) or too few detected genes (1.5-fold under median). We next determined the variable genes within each cluster of cells and performed PCA. Finally, we recovered all the cells that were removed in the last step by projecting them back into PC space.

## Pseudo-timing in PC space

For each cell cluster that showed a ring-like structure in PC1 vs. PC2 space (Clusters 3, 4, 8, 11, 12, 22, 30) (Fig. 1C), we determined its center by calculating the mean of the 5% and 95% quantiles for $x$ and $y$, separately. We then calculated the angle (from 0 to 360°) of each cell with respect to the center using the atan2 function and used that angle as a measure of relative pseudo-time. We specifically refer to the word "relative" because at this point, pseudo-time is not synchronized among the ring-like structures in principal component space. The developmental time of e.g., 20 h in *C. elegans* development might be located at 30° in one cluster but at a completely different angle in another cluster. Furthermore, the directions of the time progression are not synchronized either. In one cluster developmental time might progress clockwise while in another cluster it could be counterclockwise.

## Inference of cell-type-specific relative pseudo-time gene expression profiles

We inferred relative pseudo-time expression profiles for oscillating cell types (clusters 3, 4, 8, 11, 12, 22, 30) as follows: one gene at a time, we selected all the cells in a given cell cluster and fit a cubic smoothing spline in the pseudo-time ($x$ axis) vs. single-cell expression ($y$ axis) scatter. For this, the R function smooth.spline() was used with the smoothing parameter spar set to 0.9. To ensure continuity across the 0 and 360° boundary in the resulting expression profiles, we duplicated the data on both sides of the scatterplot, once shifted by 360° on the $x$ axis and once shifted by −360°. Finally, we resampled the resulting interpolation line at a fixed grid from 0 to 350° in increments of 10° (36 pseudo-time points). We refer to the result as a relative pseudo-time gene expression profile for individual cell types.

## Synchronization of cell-type-specific pseudo-time expression profiles

To obtain time- and tissue-resolved expression profiles in absolute time, rather than relative, we synchronized the expression profiles using an external bulk RNA-seq time course experiment as a common reference (Meeuse et al, 2020). We hypothesized that for each oscillating tissue, there should be an optimal starting point and an optimal direction in the PC1 vs. PC2 space such that the correlation to the bulk RNA-seq would reach a maximum. Hence, we first selected the genes that showed oscillatory gene expression in bulk and that were detected in our relative-time expression

profiles from the single-cell data. We then normalized the latter by dividing the expression levels for each gene (in linear space) by its maximum level in any cell type (resulting in a maximum expression of 1 for every gene). To create a comparable dataset from the bulk RNA-seq experiment, we used the previously determined phase information for every gene (from 0 to 360°) to synthesize expression profiles (using normal distributions) that peak at the experimentally determined phase (also normalized to a maximum of 1). To determine the optimal starting point and an optimal direction in the PC1 vs. PC2 space for every cell type, we created multiple versions of the expression profiles (which are in relative time), each starting at a different time point and each of those in both clockwise and counterclockwise directions. This resulted in a total of 2*36 = 72 versions, which we correlated to the bulk. We quantified the similarity using the Pearson correlation coefficient and selected the expression profile version with the greatest correlation as the absolute pseudo-time gene expression profile for any given cell type (Fig. EV1E). An indicator for successful synchronization is a situation where we see a local correlation maximum in only one direction. This was observed for the seven oscillation tissues, but not for the intestine.

### Collecting animals at molt entry and molt exit

Synchronized HW1939 (*eft-3p::luc::gfp*) L1 worms were seeded (1 animal per well) in two 384-well plates containing S-Basal (5.85 g NaCl, 1 g $K_2HPO_4$, 6 g $KH_2PO_4$, 1 mL cholesterol (5 mg/mL in ethanol) up to 1 L with $H_2O$) (Stiernagle, 2006), OP50 bacteria ($OD_{600} = 1$), and 100 μM Firefly D-Luciferin (p.j.k., 102111). Developmental trajectories were then recorded, as described (Meeuse et al, 2020; Olmedo et al, 2015) on a luminometer for 26 and 28 h so that the majority of worms were at L3 molt entry and L3 molt exit, respectively. Plates were flash frozen in liquid nitrogen and then incubated at −80 °C for ~30 min to kill the worms but prevent excess degradation. Based on luciferase traces, the wells containing worms at the appropriate developmental timepoint were identified, the plates were thawed, and individual worms were collected from their wells and pooled for RNA isolation (Direct-zol RNA microprep kit - R2062, Zymo Research), library preparation (TruSeq Illumina mRNA-seq protocol), and sequencing (HiSeq 2500 50 cycle single-end protocol).

### Calibration of oscillatory developmental trajectories

During *C. elegans* larval development, at least 3739 genes oscillate at the mRNA level (Meeuse et al, 2020), for which we know the relative peak phases (not calibrated to developmental time). This information can be used to visualize the developmental trajectory of a given RNA-seq time course experiment (Nahar et al, 2024). The method works as follows: We first split the 3739 genes into four phase-binned gene sets, gene set 1 containing genes with a peak phase of 0° ± 45° (effectively 315°–360° and 0°–45°) and gene sets 2, 3, and 4 containing genes with a peak phase of 90° ± 45°, 180° ± 45°, and 270° ± 45°, respectively. Then we calculate the average gene expression levels for those 4 gene sets in the mRNA-seq time course data set (after mean normalization at the gene level) and subtract the two anti-phase pairs (x=geneset1-geneset3, y=geneset2-geneset4) to obtain 2 orthogonal timing readouts $(x, y)$ for each RNA-seq sample. Visualized in a scatterplot (Appendix

Fig. S2), this typically results in circular trajectories with time progressing in counterclockwise orientation. In this space, we can estimate the pseudo-time of every data point $(x, y)$ by calculating the respective polar coordinate angle. To calibrate those timing trajectories with respect to larval development, we performed RNA-seq of animals at either molt entry or molt exit, obtained as described above, and used the above procedure to determine the pseudo-time of those samples. We obtained 42.5° for the molt entry sample and 125.8° for the molt exit sample. We then rotated the original peak phases (Meeuse et al, 2020) by −125.8°, which resulted in new calibrated phases for the 3739 genes oscillating genes such that 0° corresponds to molt exit (i.e., the transition to a new larval stage) and −83.2° to molt entry. To simplify the generation of calibrated (molt-sync) developmental trajectories for future experiments, we also provide calibrated phase-binned gene set assignments (available in the associated GEO submission).

### Collecting animals for ATAC-seq and RNA-seq

Synchronized L1 worms were plated at a density of 2000 animals per 10-cm NGM, 2% plates, seeded with OP50. The worms were then grown at 25 °C and collected hourly from 14 to 30 h post plating, covering the L2 and L3 stages. At each time point, 14,000 worms (7 plates) were collected for ATAC-seq and 2000 worms (1 plate) were collected for RNA-seq. Worms were washed with M9 buffer until no bacteria were visible, and then spun one final time in order to remove as much supernatant as possible. The worm pellet was then frozen in liquid nitrogen and transferred to −80 °C freezer until further use.

For GRH-1 depletion experiments, synchronized HW2434 (*grh-1::degron*) L1 worms were grown at a density of 1 worm/μL in liquid culture (S-Basal with OP50 ($OD_{600} = 1$)) for 20–21 h at 20 °C with 120 RPM shaking. Worms were collected for timepoint 0 (replicate 1 only), before the culture was split equally (380 mL each) into two new flasks and treated with either 950 μL 100% EtOH or 950 μL of 0.1 M auxin (250 μM final). Overall, 15,000 worms for ATAC-seq and 2000–5000 worms for RNA-seq were collected (as above) every hour for 10 h.

### ATAC-seq time course

The ATAC-seq protocol was performed as previously described (Daugherty et al, 2017), replacing the lysis buffer with the Omni-ATAC buffers (Corces et al, 2017). For nucleus isolation, we performed all steps on ice or at 4 °C. Samples were thawed on ice (processed one at a time) and resuspended in 150 μL of Omni-ATAC buffer (10 mM Tris-HCl pH 7.4, 10 mM NaCl, 3 mM $MgCl_2$, 0.01% digitonin, 0.1% Tween 20, 0.1% NP40). Samples were then transferred to a pre-chilled Wheaton stainless-steel Dounce Dura-Grind Tissue grinder (DWK Life Sciences 357572 - Length 114 mm, 7 mL) and dounced with 3 strokes. Slurry was transferred to a fresh 1.5-mL tube and centrifuged at 200×g for 1 min to remove worm debris. Supernatant (containing nuclei) was then transferred to a new 1.5-mL tube and set aside. The pellet was resuspended in 150 μL of Omni-ATAC buffer, and homogenization steps were repeated as above 5–7 times (until no intact worms remained). Nuclear supernatants from all homogenization steps were pooled. Pooled nuclei were then centrifuged at 200×g for 1 min to remove any remaining debris, and the supernatant was transferred to a

fresh 1.5-mL tube. Nuclei were then pelleted at $1000 \times g$ for 10 min in a swinging bucket centrifuge, and the supernatant was removed without disrupting the pellet.

For the transposition reaction, the nuclear pellet was gently resuspended in 50 μL of transposition mixture (5 μL 5× TD buffer, 2.5 μL Transposase (100 nM final), 0.5 μL 1% digitonin, 0.5 μL 10% Tween 20, 16.5 μL PBS, 21 μL ddH$_2$O). Samples were tagmented at 37 °C for 30 min on a heat block, shaking at ~600 rpm. DNA fragments were then immediately purified using the minElute PCR purification kit (Qiagen, 28004), eluting in 16 μL of elution buffer. Libraries were prepared using the NEBNext High-Fidelity PCR Master Mix (M0541S), and initially amplified with 5 PCR cycles. The number of remaining cycles was determined based on qPCR amplification of the library (as described in the original ATAC-seq protocol(Buenrostro et al, 2015)). Amplified libraries were purified using AMPure XP beads (A63881) at a 1.6× ratio and eluted in 20 μL of Elution buffer.

Libraries were subsequently sequenced using either the NextSeq HIGH-OUT 75 Cycle Paired-end protocol (N2 timecourse) or NovaSeq 6000 Paired-end protocol (*grh-1::degron* timecourse).

Note: Upon closer inspection we found that the nuclei purification protocol was suboptimal to yield nuclei from the pharynx as we could observe intact pharynges in our sample prep. Thus, it is likely that the protocol introduces a bias that causes depletion of pharynx-specific peaks.

## ATAC-seq processing

ATAC-seq data were mapped to the *C. elegans* genome (ce11) using the Align (qAlign) function from the QuasR package in R. The fragment size distributions of the individual libraries showed multiple peaks, the short fraction (<120 bp) corresponding to transcription factors footprints and long fraction (>120 bp) to nucleosomes. We thus considered fragments with a read length of less than 120 bp for further analysis. To obtain ATAC-seq peaks we combined the alignments of all samples and ran MACS2 peak finder. We quantified the number of reads in all the peaks using the function(qCount() in the package QuasR. We further performed quantile normalization to correct for nonlinear trends in the data, followed by log$_2$ transformation with pseudocount of 8 ($\log_2(x + 8)$) for the N2 timecourse. An increased pseudocount of 32 ($\log_2(x + 32)$) was used for the *grh-1::degron* timecourse to further reduce noise, which was necessary in this case due to the downstream analysis of individual timepoints as opposed to the N2 timecourse, where cosine curves were fit on all timepoints simultaneously. In the first replicate of the *grh-1::degron* timecourse, we noticed two outlier samples, EtOH_24h and Auxin_25h, which showed lower correlation to all other samples but high correlation to each other. These two samples were processed simultaneously, suggesting a technical batch effect. We corrected for this batch effect by first calculating the average ATAC-seq intensity profile of those two samples and then subtracting the average profile of the neighboring four samples in the timecourse. This resulted in a batch correction vector, which we then subtracted from the two outlier samples to obtain corrected intensities for those two samples.

## RNA-seq time course

RNA was isolated by crushing worms (in 700 μl TRI Reagent (MRC Inc.)) in a MP Biomedical FastPrep-24 Bead Beating Grinder for five rounds (time = 25 s; speed = 8.0 m/s). Subsequently, worms were spun for 20 min at 4 °C to remove debris and supernatant was applied to columns of the RNA extraction kit (R2062, Zymo Research) and steps were followed as in the protocol. DNase treatment was carried out on the column for 15 min at room temperature. RNA was eluted in 15–20 μL of water. RNA libraries were prepared according to the TruSeq Illumina mRNA-seq (stranded—high input) protocol (N2 timecourse) and Illumina Stranded mRNA-seq protocol (*grh-1::degron* timecourse). Subsequently, the library was sequenced using Hiseq 50 Cycle Single-end reads (N2 timecourse) and NovaSeq 6000 Paired-end reads (*grh-1::degron* timecourse).

## RNA-seq processing

RNA-seq data was mapped to the *C. elegans* genome (ce11) using the Align (qAlign) function from the QuasR package in R with splicedAlignement = TRUE, using the aligner HISAT2 including an exon-exon junction database. Gene expression levels were quantified using the qCount() function. For annotations, coding transcript info from WormBase WS270 was used. We further compensated for differences in library sizes by scaling each library to the average library size, and log$_2$-transformed the data using a pseudocount of 8 ($\log_2(x + 8)$). To visualize developmental trajectories, we applied the orthogonal gene set approach using calibrated phase-binned gene set assignments (see Calibration of oscillatory developmental trajectories). We then estimated the wrapped pseudo-time of every data point (x,y) by calculating the respective polar coordinate angle. We then unwrapped the angle values and applied a LOESS fit to obtain the final inferred unwrapped pseudo-time for each sample.

## Cosine curve fitting on the ATAC-seq data

To study potential oscillations in the ATAC-seq data, we performed cosine curve fitting separately for every ATAC-seq peak using a linear model. In addition to the two oscillatory predictors $\cos(\varphi_s)$ and $\sin(\varphi_s)$ where s denotes the sample, we also included a linear term $\lambda_s$ to capture graded changes in intensity. For $\varphi_s$, we directly used the inferred unwrapped pseudo-times (in radians) obtained from the matching RNA-seq experiments (see above). This allowed us to compensate for slight differences in the timing between the two replicates (Fig. EV2D), to combine the two replicates in one single cosine curve fitting procedure and to obtain calibrated phases (molt-sync, see "Methods"). ATAC-seq peaks with an amplitude >0.25 were classified as oscillating, and ATAC-seq peaks with an amplitude <0.25 and an absolute graded coefficient >0.25 were defined as graded, all others were designated as flat.

## Cosine curve fitting on the RNA-seq data

To quantify oscillatory gene expression, we performed cosine curve fitting the same way as for the ATAC-seq data, but replacing the linear component $\lambda_s$ by the average expression of "rising genes" as defined in (Hendriks et al, 2014). We did so to account for the sudden increase in

expression of over a thousand genes at 24 h (Hendriks et al, 2014), a trend that is only poorly captured by a linear term.

## Single worm imaging and analysis

The imaging protocol was performed as described previously (Meeuse et al, 2020) with minor changes. Briefly, microchambers were made using 4.5% agarose, and filled with OP50 bacteria. Embryos were collected from plates containing gravid adults and were seeded into wells (1 embryo/well). Microchambers were then transferred, upside down, to a microscope slide (Ibidi – μ-slide 2 Well ibiTreat 80286), and sealed with low-melting agarose (3% in S-Basal) to avoid detachment and drifting. The slide was then closed and sealed with parafilm to avoid dehydration. Chambers were imaged using a Yokogawa CSU W1-T2 spinning disk confocal scanning unit with a ×20 air objective. The worms were grown at 22 °C and imaged every 10 min over 60 h. At each time point brightfield and fluorescent images were acquired in parallel for 25 stages in z (2-μm step size). Brightfield images were acquired with an exposure time of 10 ms, and simultaneous mCherry and GFP were acquired with an exposure time of 50 ms.

Image processing was performed using a Python-based workflow. First, the worms were automatically detected and segmented in the chamber based on the mCherry signal in 3D using a self-developed adaptive masking function. For each timepoint and each chamber, average GFP intensities were calculated over the segmented volume and saved for further post processing (using the regionprops function from skimage (van der Walt et al, 2014)). Based on a linear regression method, the background signal/autofluorescent signal in the worm was predicted using worms that do not express GFP. The predicted background signal was then subtracted from the quantified GFP intensity. A pseudocount of ~50 was included due to noisy signal in early timepoints which sometimes resulted in negative values after background correction. Data was then smoothed by applying a Savitzky–Golay filter.

For each animal and each larval stage, molt exit was annotated visually based on shedding of the cuticle in brightfield images. Processed individual GFP traces could then be aligned to the respective larval stages, with time normalized for average duration.

## NHR-23 ChIP-seq

ChIP-seq was performed with endogenously tagged *nhr-23* (JDW29—*nhr-23::GFP::degron::3xFLAG*) (Ragle et al, 2020). Sample preparation was performed as previously described (Meeuse et al, 2023). In brief, samples were prepared by plating 5 million synchronized L1 larvae on NA22-containing peptone-rich XL plates for 20 h at 25 °C. Worms were collected in PBS supplemented with protease inhibitors (1 mM PMSF + 1 tablet cOmplete EDTA-free (Roche—11873580001) per 50 ml PBS). "Popcorn" was created by dripping resuspended larvae into liquid nitrogen and subsequently ground into a fine powder. Samples were then crosslinked with 1.1% formaldehyde in PBS + protease inhibitors for 20 min on a rotating wheel, before being quenched with 125 mM glycine for 5 min, rotating at room temperature. Chromatin was then sonicated with a Diagnode Biorupter Pico at 4 °C (30 cycles, 30 s on/30 s off). Chromatin concentration was measured via Nano-Drop, and 100 μg of chromatin was used for each immunoprecipitation. Chromatin was added to 50 μL of protein G

Dynabeads (ThermoFisher Scientific - 10003D) pre-incubated with 5 μg of anti-GFP antibody (Abcam—ab290), and incubated overnight at 4 °C with rotation. The beads were washed, the eluate was treated with Proteinase K and RNase A and crosslinks were reversed overnight at 65 °C before purifying samples using the ChIP DNA Clean and Concentrator Kit (Zymo—D5205). Sequencing libraries were prepared using the ChIP-seq NEB Ultra protocol (New England Biolabs) and sequenced using the NovaSeq 6000 Paired-end protocol.

## Processing of ChIP-seq data

For the quantitative assessment of the binding patterns of nine molting clock transcription factors, we performed ChIP-seq for NHR-23 (see above) and downloaded the raw sequencing data (IP and respective Input) for the factors BLMP-1, LIN-14, NHR-43, NHR-21, HLH-11, NHR-25, and HAM-2 from https://epic.gs.washington.edu/modERN/. Samples with multiple replicates were combined. ChIP-seq data for GRH-1 was downloaded from GEO (Input:GSM6588392 IP:GSM6588394 from the series GSE213510). The reads were aligned to the *C. elegans* genome (ce11) using the qAlign function from the QuasR package in R with default parameters. We quantified the number of reads in all ATAC-seq peaks (extended by 100 bp on both sides) using the function qCount from the QuasR package, setting the shift parameter to 100 (moving the read alignment positions in a strand-dependent fashion towards the fragment center). For each factor separately, we normalized the respective IP and input samples by dividing each sample by the total number of reads and multiplying by the average library size, and subsequently converting to $\log_2$ space using a pseudocount of 8. During visual data inspection, we noticed 39 ATAC-seq peaks (out of 42 K peaks) with very high $\log_2$ counts in the input samples (>11 in any input sample), which we removed. We calculated $\log_2$ ChIP enrichment values for each factor by subtracting the input sample from the corresponding ChIP sample. Assuming that the majority of ATAC-seq peaks are not bound by a given factor, we centered the $\log_2$ ChIP enrichment values for each factor around the maximum position of the $\log_2$ ChIP enrichment distribution. To determine the position of the maximum, we used the R function density with a bandwidth of 0.04 to obtain a density estimate. During quality control, we noticed one ATAC-seq peak (ce_ATAC_peaks_peak_19961) that showed an extreme $\log_2$ IP enrichment value of 5.3 for NHR-25. We inspected the alignments at that locus (chrIII:12,424,000–12,429,000) and found three 1 kb long segments in the 3 IP samples that showed uniform read amplification within each block. As this did not look like a trustworthy NHR-25 binding site, we set the nhr-25 $\log_2$ IP enrichment of that single ATAC-seq peak to 0. Generally, ChIP enrichment values from different factors are not directly comparable. Nevertheless, to obtain enrichment values that operate on a numerically similar range, we determined the 99th percentile for each factor, divided the enrichment values by the respective 99th percentile, and multiplied by the average 99th percentile for all factors. Note that this final normalization step has no impact on the performance of the various linear models that are fit throughout the manuscript, as this step only involves a multiplication of each predictor by a factor-specific constant.

## Randomization procedure for the predicted sum of TF vectors

For each ATAC-seq peak, we can predict the combined impact of all nine TFs (prediction vector) by multiplying the activity vector of each TF ($x'$, $y'$) by its ChIP enrichment value in the given ATAC-seq peak, and then summing up all those 9 vectors. To generate a randomized control, we considered two options, randomization of the ChIP enrichment matrix or randomization of the TF activity vectors. The ChIP enrichment matrix has a complex underlying structure (extensive co-binding), making it difficult to randomize it such that the underlying structure is still preserved. We thus chose to randomize the TF activity vectors. First, we rearranged our linear model such that the TF activity vectors have a length of one by first dividing each TF activity vector by its length (normalized activity vectors) and at the same time multiplying the TF enrichment values of a given TF by the respective TF activity vector length (normalized TF enrichments). Note that this operation is just a mathematical rearrangement and does not impact the output of the model in any way. For the randomization procedure, during the prediction step, where the normalized activity vectors are multiplied by the normalized TF enrichments and summed up, we permuted the angles of the normalized activity vectors (e.g., replacing the angle of NHR-25 with that from BLMP-1), one ATAC-seq peak at a time (Altun and David (2024)).

## Data availability

All raw datasets generated for this manuscript, as well as certain processed tables, were deposited in the Gene Expression Omnibus (GEO) as a superseries with the accession code GSE288914, available at https://www.ncbi.nlm.nih.gov/geo/query/acc.cgi?acc=GSE288914. Code associated with this manuscript can be downloaded from: https://github.com/fmi-basel/ggrossha-ce-chromatin-oscillations-TF-modelling.

The source data of this paper are collected in the following database record: biostudies:S-SCDT-10_1038-S44320-025-00155-9.

## Peer review information

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

## Acknowledgements

We thank L Morales Moya, AAT Smith, and P Papasaikas for discussions and advice. We thank A Walczak, L Xu and I Katic for support with *C. elegans* transgenesis. We thank L Plantard and L Gelman for support with imaging. Strains were kindly provided by J Ward and by the *Caenorhabditis* Genetics Center (CGC), funded by NIH Office of Research Infrastructure Programs (grant no. P40 OD010440). We thank B Towbin, M Bühler, and L Giorgetti for their critical comments on manuscript drafts. SPM was financed with an Ambizione grant evaluated by the Swiss National Science Foundation (PZ00P3_174153). This work is part of a project that has received funding from the European Research Council (ERC) under the European Union's Horizon 2020 research and innovation programme (Grant agreement No. 741269, to HG) and from the Swiss National Science Foundation (#310030_207470, to HG). MWMM and GB received support from Boehringer Ingelheim Fonds PhD fellowships, SN from a European Union's Horizon 2020 Marie Skłodowska-Curie Actions Postdoctoral fellowship (#842386). The FMI is core-funded by the Novartis Research Foundation.

## Author contributions

**Dimos Gaidatzis**: Conceptualization; Software; Formal analysis; Investigation; Visualization; Methodology; Writing—original draft. **Maike Graf-Landua**:

Formal analysis; Investigation; Methodology; Writing—original draft. **Stephen P Methot**: Supervision; Investigation; Methodology; Writing—original draft. **Michaela Wölk**: Investigation. **Giovanna Brancati**: Methodology. **Yannick P Hauser**: Resources; Formal analysis; Investigation. **Milou Meeuse**: Investigation. **Smita Nahar**: Investigation. **Kathrin Braun**: Investigation; Methodology. **Marit van der Does**: Formal analysis. **Sirisha Aluri**: Investigation. **Hubertus Kohler**: Investigation. **Sebastien Smallwood**: Methodology; Project administration. **Helge Großhans**: Conceptualization; Supervision; Funding acquisition; Writing—original draft; Project administration.

Source data underlying figure panels in this paper may have individual authorship assigned. Where available, figure panel/source data authorship is listed in the following database record: biostudies:S-SCDT-10_1038-S44320-025-00155-9.

## Disclosure and competing interests statement

The authors declare no competing interests.

# Expanded View Figures

**Figure EV1.   Assigning and pseudo-timing oscillating tissues based on scRNA-seq.**                                              ▶

(A) Correlation ($R^2$) of gene expression profiles from the cell clusters (rows) against the *C. elegans* tissues identified in (Cao et al, 2017). Clusters were assigned to the tissue with the highest correlation, except for cluster 10, which had a low correlation to multiple tissues and remained unassigned. (B) The expression of annotated oscillating genes (Meeuse et al, 2020) in all the identified tissues. Genes are ordered by 1D-tSNE. Left, scatterplot for the molt-synchronized phase of expression (degrees) of oscillating genes, indicated by the position of the dot along the *x* axis. Right, heatmap of expression levels for each oscillating gene (rows) in individual tissues (columns). (C) PCA plot comparing PC1 versus PC2 for seam cells, separating the cells into four panels according to their experimental timepoint. (D) Determining a tissue-specific and time-resolved expression profile for an example gene. A cubic smoothing spline (red circles) was fit for the gene *Y54G2A.76* on the scatterplot comparing the assigned pseudo-time, in PC space, versus normalized UMI counts per cell for each cell (black dots). Expression data shown is from the seam cell cluster. (E) Synchronization of the tissue and time-resolved expression profiles by comparison to an external bulk RNA-seq reference. For each tissue, we created variant expression profiles starting at different positions within the cycle and considering both clockwise (red) and counterclockwise (black) orientations. The panels display the Pearson correlation coefficients between the variant expression profiles and the bulk RNA-seq reference. (F) Correlation between experimental time and inferred pseudo-time. For each experimental timepoint, we determined the cell density along pseudo-time accounting for the calibration step illustrated in (E). The heatmaps show $\log_2$ cell density enrichments.

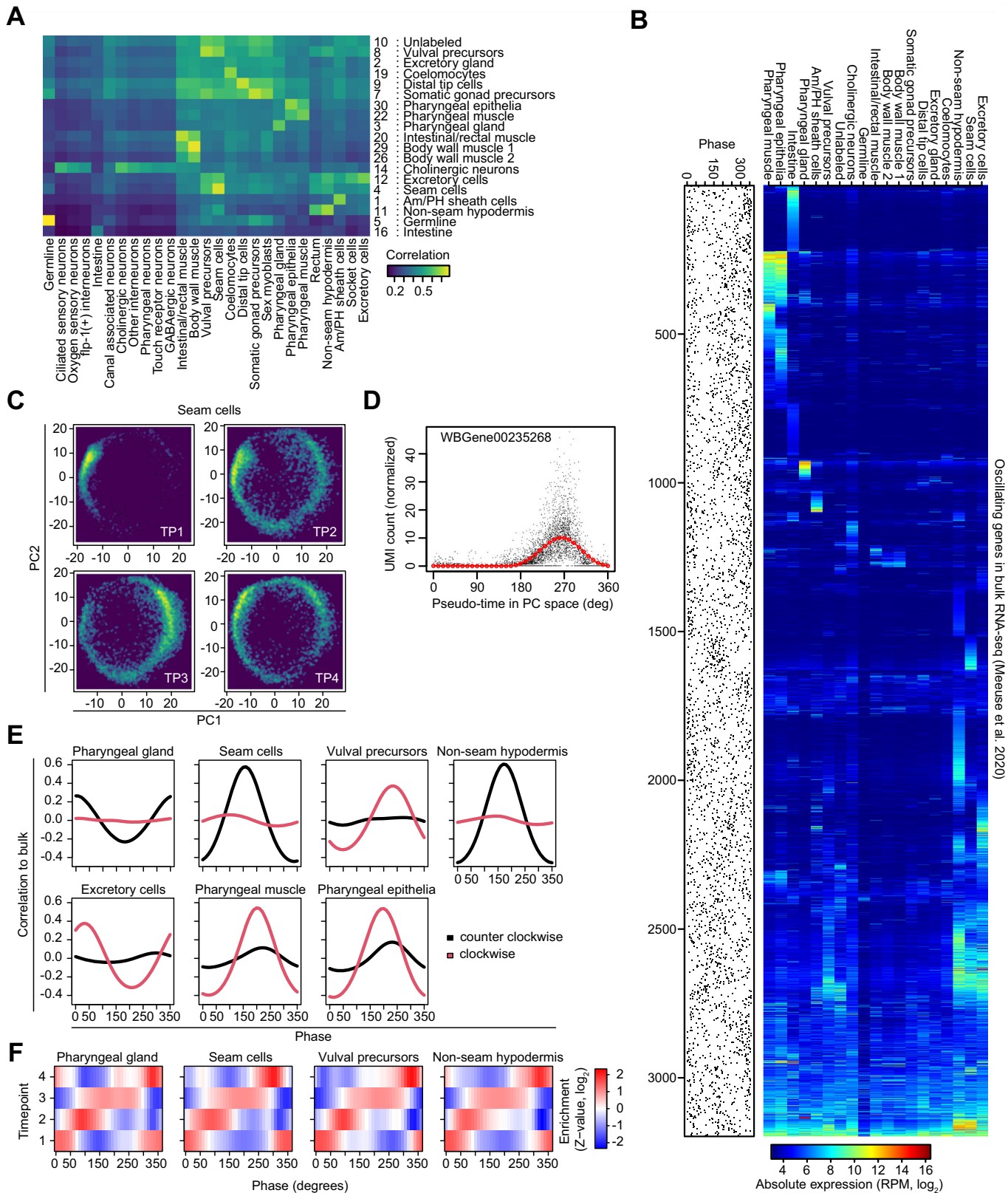

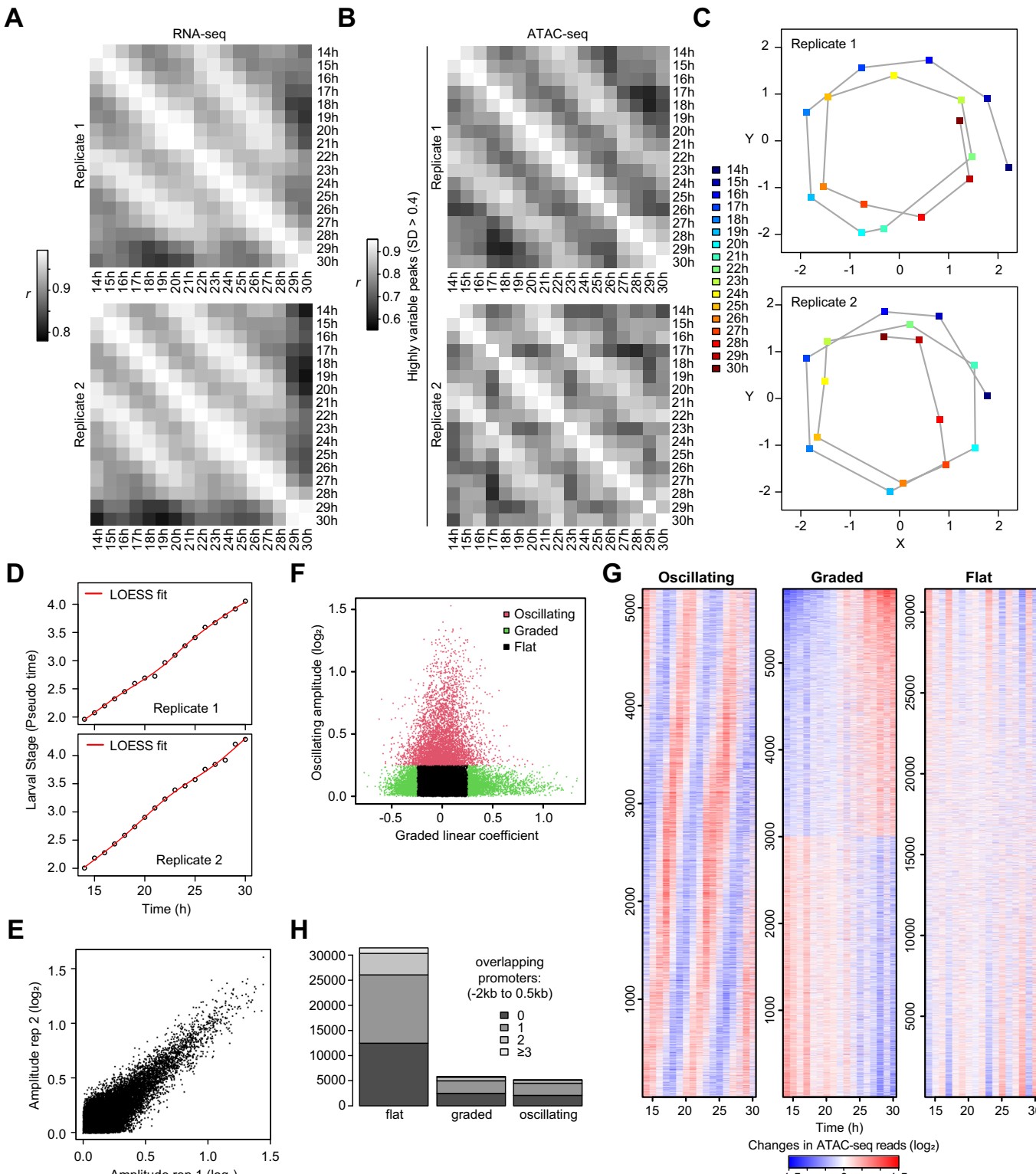

◀ **Figure EV2.   Matched RNA-seq and ATAC-seq larval development time course.**

(A, B) Pairwise correlation heatmap for $\log_2$-transformed RNA-seq (A) or ATAC-seq (B) data in each of the 17 time points from the 2 independent replicates. Periodic changes in correlation indicate oscillations in both RNA-seq and ATAC-seq. r indicates the Pearson correlation coefficient. In (B), only highly variable peaks are used (SD > 0.4). (C) Scatter plot visualizing the developmental trajectories for the 2 replicate RNA-seq time course datasets. Trajectories were calculated similar to Appendix Fig. S2 (see "Methods"). (D) Scatter plot comparing the experimental time (x axis) to the unwrapped pseudo-time (y axis) for the 2 replicate RNA-seq time course datasets. We transformed the trajectories from panel (C) into polar coordinates to calculate the wrapped pseudo-time, which was unwrapped to obtain the final pseudo-time (y axis). A LOESS fit (red line) was used to smooth the pseudo-time. (E) Scatterplot comparing ATAC-seq amplitudes from replicate 1 versus replicate 2 for all peaks. Amplitudes were obtained by performing cosine fits on each replicate separately. (F) Cosine fitting of ATAC-seq peaks was used to identify their amplitude, which was plotted against the trend derived using a linear regression (trend is the graded linear coefficient). This reveals three distinct classes; graded (green), flat (black) and oscillating (red). (G) Heatmaps of mean-normalized and $\log_2$-transformed changes in ATAC-seq reads at individual peaks over time. Peaks were classified according to their amplitude and graded linear coefficient (F). Oscillating peaks are sorted by their peak-phase, graded peaks by the graded component, and flat peaks by principal component 1. See Fig. 2A for replicate experiment. (H) Barplot of the total number of ATAC-seq peaks from each category that overlap the indicated number of promoters.

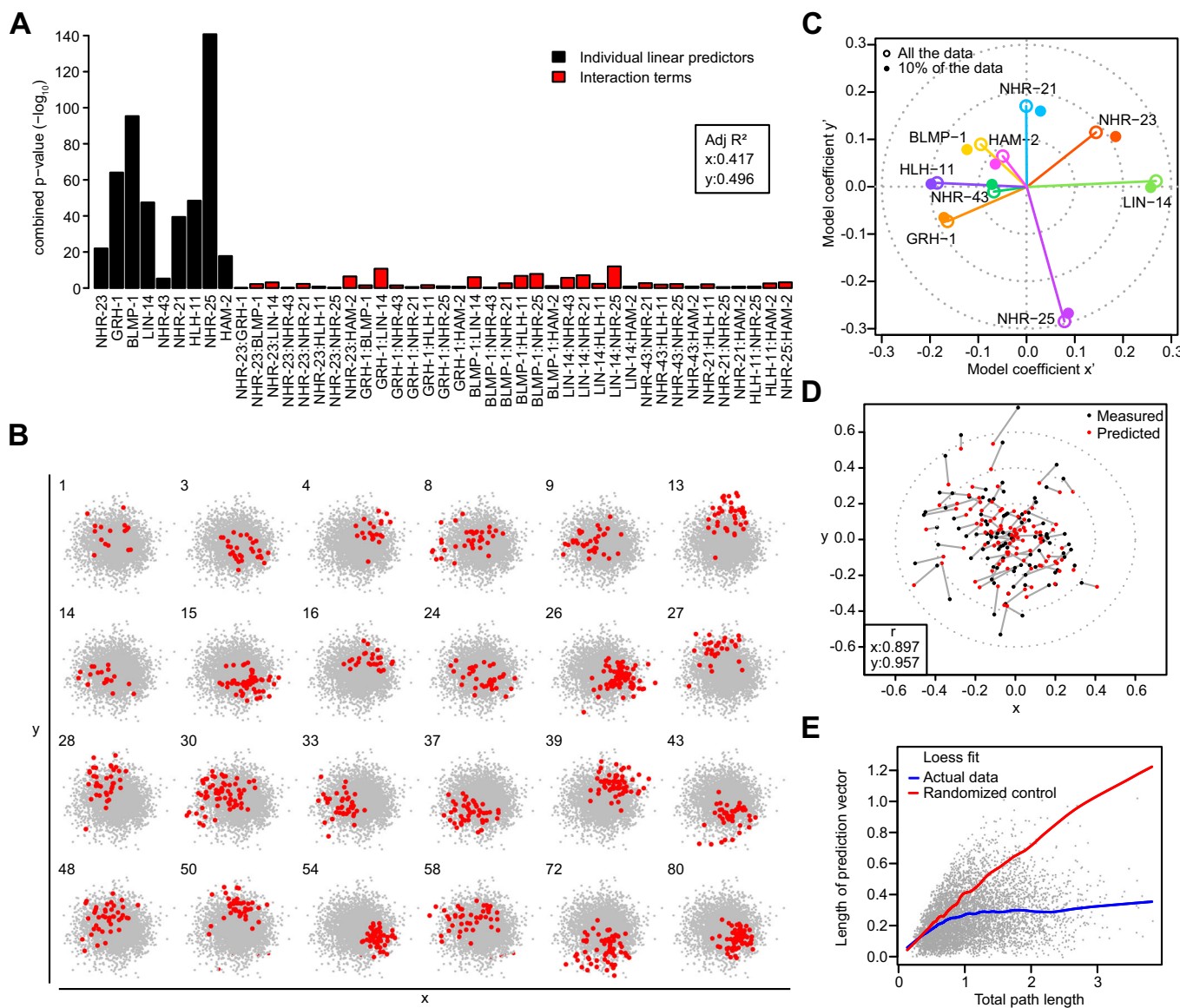

**Figure EV3. Binding of 9 transcription factors is predictive of ATAC-seq phase and amplitude.**

(A) Barplot indicating the relative influence of each molting clock TF on the linear model including pairwise interaction terms. Black bars indicate independent contributions, red bars contributions through interactions. The predictive power of the model (adjusted $R^2$) is indicated on the right. (B) Radar plots representing clusters depicted in Fig. 5C, generated based on ChIP-seq enrichments of molting clock TFs (cluster numbers are indicated in the top left corner). Each plot displays peak phase and amplitude (as x,y Cartesian coordinates) for all ATAC-seq peaks (gray dots), with cluster-specific peaks highlighted (red dots). (C) Representation of the output from a linear model (similar to Fig. 4E), generated using either 10% of ATAC-seq peaks (filled circles) or all peaks (empty circles). (D) Scatterplot comparing predicted and measured vectors for all 100 clusters from Fig. 5B. The Pearson correlation coefficient (r) for both x and y predictions respectively are indicated in the bottom left corner. (E) Scatterplot comparing the total path length (generated by summing up the absolute lengths of the individual phase-vectors for each TF) against the predicted vector length (generated after summing the phase-vector contributions of all TFs) for each ATAC-seq peak. This was done for all ATAC-seq peaks used in the modeling. A LOESS fit was applied to the data and is plotted (blue line). A control was generated by randomly combining the TF enrichments from all ATAC-seq peaks, and a second LOESS fit was applied (red line).

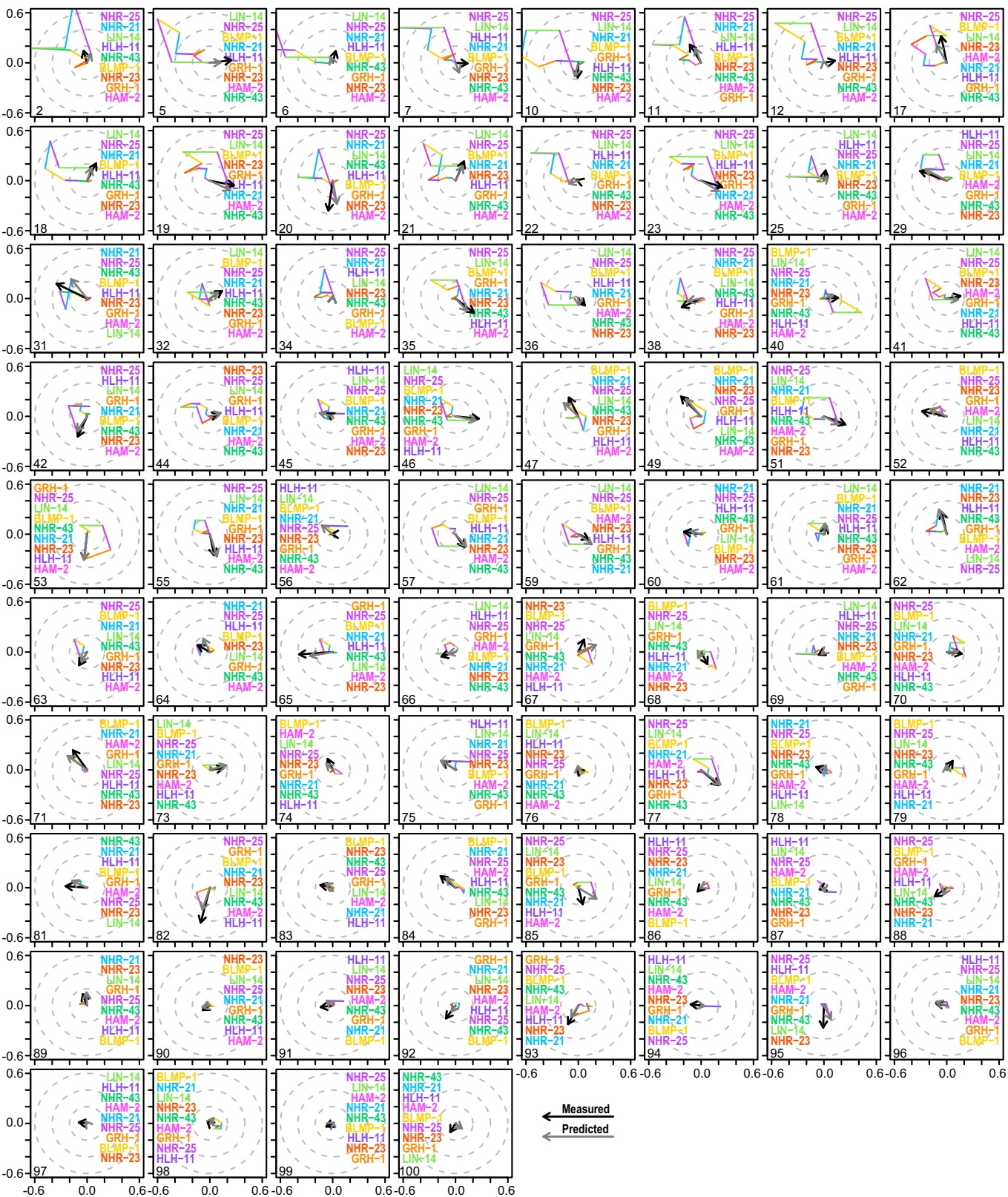

◀  **Figure EV4. Phase-vector predictions for remaining ATAC-seq clusters.**

Phase-vector predictions for individual ATAC-seq clusters (gray arrows) were generated by adding the phase-vector contributions (colored lines – average TF enrichment*x/y coefficients) of each TF. The average measured output vector for the cluster is indicated (black arrow). For each cluster, the phase-vectors are added together in ascending order, starting with the shortest phase, while the list of TFs is organized in descending order. Clusters are ordered based on their total path length (generated by adding the absolute length of each TF vector), aligned left to right, top to bottom. The 76 clusters not included in Fig. 5C are shown here.

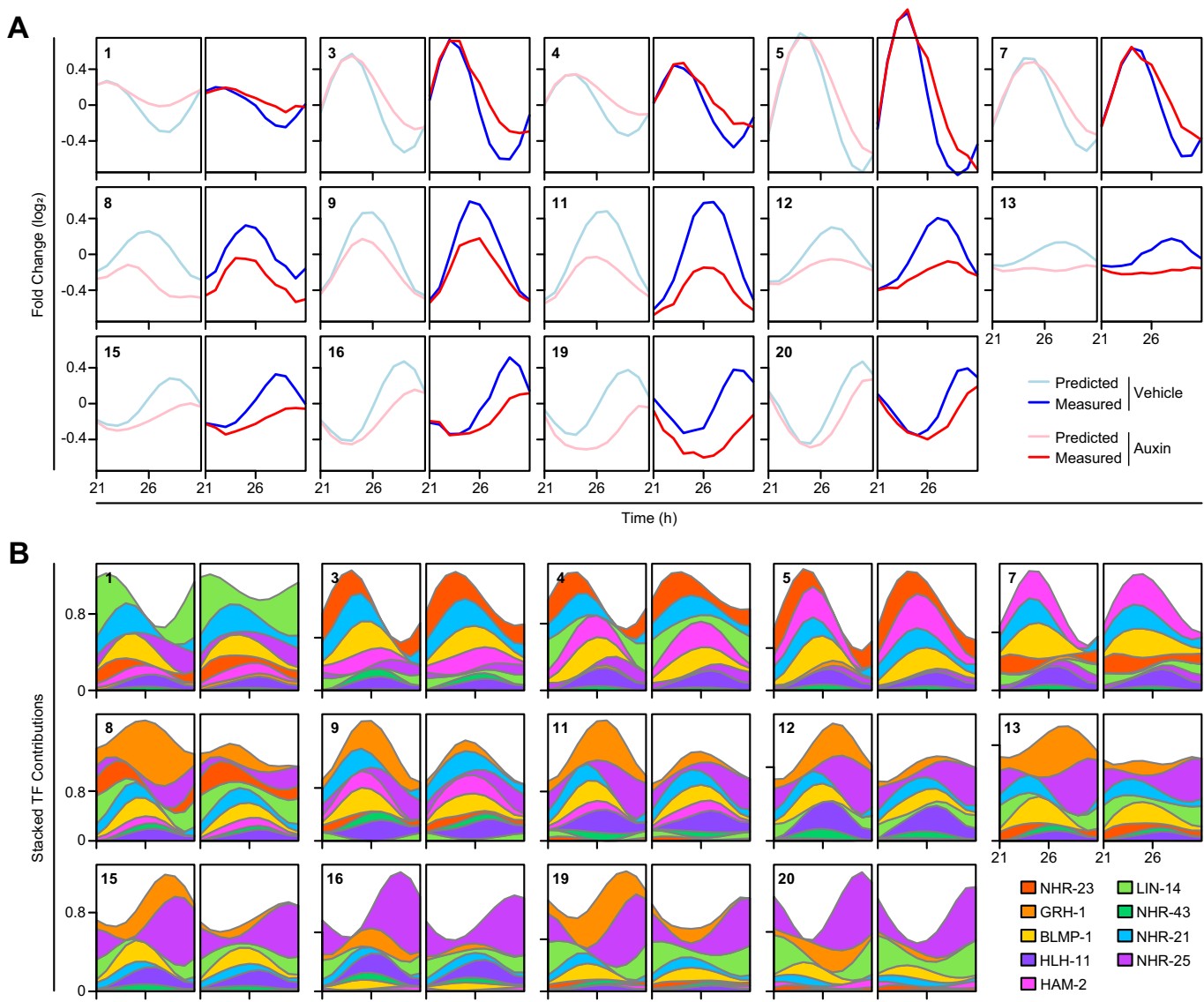

**Figure EV5. Differential ATAC-seq predictions upon GRH-1 depletion.**

(A, B) Depict clusters from Fig. 6C that were not highlighted in Fig. 6D,E and had more than 10 peaks. (A) Average predicted (left) or measured (right) accessibility changes over time for each cluster of differentially accessible peaks, comparing vehicle and auxin treatments. (B) Plots compiling individual TF contributions that generate the predicted chromatin accessibility over time. TFs are sorted by maximal activity difference between the vehicle and auxin conditions.

