## [Peer Review File · Molecular Systems Biology]

A scheduler for rhythmic gene expression

Dimos Gaidatzis, Maike Graf-Landua, Stephen Methot, Michaela Wölk, Giovanna Brancati, Yannick Hauser, Milou Meeuse, Smita Nahar, Kathrin Braun, Marit van der Does, Sirisha Aluri, Hubertus Kohler, Sebastien Smallwood, and Helge Großhans

Corresponding author(s): Helge Großhans (helge.grosshans@fmi.ch) , Dimos Gaidatzis (Dimosthenis.Gaidatzis@fmi.ch)

Review Timeline:	Submission Date:	23rd Apr 25
	Editorial Decision:	28th May 25
	Revision Received:	13th Aug 25
	Editorial Decision:	9th Sep 25
	Revision Received:	12th Sep 25
	Accepted:	19th Sep 25

Editor: Jingyi Hou

Transaction Report:

28th May 2025

Manuscript Number: MSB-2025-13069
Title: A scheduler for rhythmic gene expression
Author: Dimos Gaidatzis
Maike Graf-Landua
Stephen Methot
Michaela Wölk
Giovanna Brancati
Yannick Hauser
Milou Meeuse
Smita Nahar
Kathrin Braun
Marit van der Does
Sirisha Aluri
Hubertus Kohler
Sebastien Smallwood
Helge Großhans

Dear Helge,

Thank you again for submitting your work to Molecular Systems Biology. We have now heard back from the three reviewers who agreed to evaluate your manuscript. As you will see from the reports below, the reviewers find the study interesting, important and comprehensive. However, they raised a series of concerns, which we would ask you to address in a major revision.

I think the reviewers' recommendations are clear, so it is unnecessary to reiterate them in detail here. Particular attention should be given to Reviewer #2's concerns regarding the scRNA-seq analyses. While additional experimental validations suggested by Reviewer #3 (points #1 and #2) could strengthen the study, we do not consider them to be essential for the acceptance of the manuscript.

All other issues raised by the reviewers need to be satisfactorily addressed as well. As you may already know, our editorial policy allows in principle a single round of major revision, so it is essential to provide responses to the reviewers' comments that are as complete as possible. Please feel free to contact me in case you would like to discuss in further detail any of the issues raised by the reviewers.

On a more editorial level, we would ask you to address the following issues:

- Please provide a .docx formatted version of the manuscript text (including legends for main figures, EV figures and tables). Please make sure that the changes are highlighted to be clearly visible.
- Please provide individual production quality figure files as .eps, .tif, .jpg (one file per figure).
- Please provide a .docx formatted letter INCLUDING the reviewers' reports and your detailed point-by-point responses to their comments. As part of the EMBO Press transparent editorial process, the point-by-point response is part of the Review Process File (RPF), which will be published alongside your paper.
- Please note that all corresponding authors are required to supply an ORCID ID for their name upon submission of a revised manuscript.
- We replaced Supplementary Information with Expanded View (EV) Figures and Tables that are collapsible/expandable online (see examples in <http://msb.embopress.org/content/11/6/812>). A maximum of 5 EV Figures can be typeset. EV Figures should be cited as 'Figure EV1, Figure EV2' etc... in the text and their respective legends should be included in the main text after the legends of regular figures.

Additional Tables/Datasets should be labeled and referred to as Table EV1, Dataset EV1, etc. Legends have to be provided in a separate tab in case of .xls files. Alternatively, the legend can be supplied as a separate text file (README) and zipped together with the Table/Dataset file.

For the figures and tables that you do NOT wish to display as Expanded View figures, they should be bundled together with their legends in a single PDF file called *Appendix*, which should start with a short Table of Content. Each legend should be below the corresponding Figure/Table in the Appendix. Appendix figures and tables should be referred to in the main text as:

"Appendix Figure S1, Appendix Figure S2, Appendix Table S1" etc. See detailed instructions regarding expanded view here: <https://www.embopress.org/page/journal/17444292/authorguide#expandedview>.

-Before submitting your revision, primary datasets (and computer code, where appropriate) produced in this study need to be deposited in an appropriate public database (see [http://msb.embopress.org/authorguide - dataavailability](http://msb.embopress.org/authorguide-dataavailability) <https://www.embopress.org/page/journal/17444292/authorguide#dataavailability>).

The accession numbers and database should be listed in a formal "Data Availability" section (placed after Materials & Method) that follows the model below (see also <https://www.embopress.org/page/journal/17444292/authorguide#dataavailability>). Please note that the Data Availability Section is restricted to new primary data that are part of this study.

Data availability

- RNA-Seq data: Gene Expression Omnibus GSE46843 (<https://www.ncbi.nlm.nih.gov/geo/query/acc.cgi?acc=GSE46843>)

- [data type]: [name of the resource] [accession number/identifier/doi] ([URL or identifiers.org/DATABASE:ACCESSION])

-At EMBO Press we ask authors to provide source data for the main figures. Our source data coordinator will contact you to discuss which figure panels we would need source data for and will also provide you with helpful tips on how to upload and organize the files.

- Our journal encourages inclusion of *data citations in the reference list* to directly cite datasets that were re-used and obtained from public databases. Data citations in the article text are distinct from normal bibliographical citations and should directly link to the database records from which the data can be accessed. In the main text, data citations are formatted as follows: "Data ref: Smith et al, 2001". In the Reference list, data citations must be labeled with "[DATASET]". A data reference must provide the database name, accession number/identifiers and a resolvable link to the landing page from which the data can be accessed at the end of the reference. Further instructions are available at .

- We updated our journal's competing interests policy in January 2022 and request authors to consider both actual and perceived competing interests. Please review the policy <https://www.embopress.org/competing-interests> and update your competing interests if necessary.

Please use the heading "Disclosure statement and competing interests".

- All Materials and Methods need to be described in the main text using our 'Structured Methods' format. According to this format, the Methods section includes a Reagents and Tools Table (listing key reagents, experimental models, software and relevant equipment and including their sources and relevant identifiers) followed by a Methods and Protocols section describing the methods, ideally using a step-by-step protocol format. The aim is to facilitate adoption of the methodologies across labs.

Please download and fill our Reagents and Tools Table template (.docx), which you can find in our author guidelines: <https://www.embopress.org/page/journal/17444292/authorguide#structuredmethods>.

-Regarding data quantification:

Please ensure to specify the name of the statistical test used to generate error bars and P values, the number (n) of independent experiments (please specify technical or biological replicates) underlying each data point and the test used to calculate p-values in each figure legend. Discussion of statistical methodology can be reported in the materials and methods section, but figure legends should contain a basic description of n, P and the test applied.

Graphs must include a description of the bars and the error bars (s.d., s.e.m.).

- Please provide a "standfirst text" summarizing the study in one or two sentences (approximately 250 characters, including space), three to four "bullet points" highlighting the main findings and a "synopsis image" (550px width and 400-600 px height, PNG format) to highlight the paper on our homepage.

Here are a couple of examples:

<https://www.embopress.org/doi/10.15252/msb.20199356>

<https://www.embopress.org/doi/10.15252/msb.20209475>

<https://www.embopress.org/doi/10.15252/msb.209495>

When you resubmit your manuscript, please download our CHECKLIST (<https://www.embopress.org/pb-assets/embosite/EMBO%20Press%20Author%20Checklist-1642513524327.xlsx>) and include the completed form in your submission.

Please note that the Author Checklist will be published alongside the paper as part of the transparent process (<https://www.embopress.org/page/journal/17444292/authorguide#transparentprocess>).

If you feel you can satisfactorily deal with these points and those listed by the referees, you may wish to submit a revised version of your manuscript. Please attach a covering letter giving details of the way in which you have handled each of the points raised by the referees. A revised manuscript will be once again subject to review and you probably understand that we can give you no guarantee at this stage that the eventual outcome will be favorable.

I look forward to receiving the revised manuscript soon.

Kind regards,
Jingyi

Jingyi Hou, PhD
Senior Editor
Molecular Systems Biology

We realize that it is difficult to revise to a specific deadline. In the interest of protecting the conceptual advance provided by the work, we recommend a revision within 3 months (26th Aug 2025). Please discuss the revision progress ahead of this time with the editor if you require more time to complete the revisions. Use the link below to submit your revision:

IMPORTANT: When you send your revision, we will require the following items:

1. the manuscript text in LaTeX, RTF or MS Word format
2. a letter with a detailed description of the changes made in response to the referees. Please specify clearly the exact places in the text (pages and paragraphs) where each change has been made in response to each specific comment given
3. three to four 'bullet points' highlighting the main findings of your study
4. a short 'blurb' text summarizing in two sentences the study (max. 250 characters)
5. a 'thumbnail image' (550px width and max 400px height, Illustrator, PowerPoint or jpeg format), which can be used as 'visual title' for the synopsis section of your paper.
6. Please include an author contributions statement after the Acknowledgements section (see <https://www.embopress.org/page/journal/17444292/authorguide>)
7. Please complete the CHECKLIST available at (<https://bit.ly/EMBOPressAuthorChecklist>).

Please note that the Author Checklist will be published alongside the paper as part of the transparent process (<https://www.embopress.org/page/journal/17444292/authorguide#transparentprocess>).

See also figure legend guidelines: <https://www.embopress.org/page/journal/17444292/authorguide#figureformat>

9. Please note that corresponding authors are required to supply an ORCID ID for their name upon submission of a revised manuscript (EMBO Press signed a joint statement to encourage ORCID adoption).

(<https://www.embopress.org/page/journal/17444292/authorguide#editorialprocess>)

Currently, our records indicate that the ORCID for your account is 0000-0002-8169-6905.

Link Not Available

11. Include a Reagents and Tools Table, which can be downloaded from our author guidelines

(<https://www.embopress.org/page/journal/17444292/authorguide#structuredmethods>)

*** PLEASE NOTE *** As part of the EMBO Press transparent editorial process initiative (see our Editorial at

<https://dx.doi.org/10.1038/msb.2010.72>), Molecular Systems Biology publishes online a Review Process File with each accepted manuscripts. This file will be published in conjunction with your paper and will include the anonymous referee reports, your point-by-point response and all pertinent correspondence relating to the manuscript. If you do NOT want this File to be published, please inform the editorial office at contact@molsystbiol.org within 14 days upon receipt of the present letter.

Reviewer #1:

Gaidatzis and coworkers report a very detailed and carefully executed study on oscillatory gene expression in the nematode *Caenorhabditis elegans*. Rhythmic gene expression is observed in multiple biological systems and play critical roles during development, tissue homeostasis, etc. yet underlying genetic mechanisms are unclear. Here, the authors determined gene expression profiles in individual cells from synchronised *C. elegans* larvae at four different time points. They identified seven cell types that showed oscillatory gene expression. By calculation of pseudo-time gene expression profiles and comparing these with bulk RNA-seq data, the amplitude and phase for each gene were determined. Next, samples were prepared at 1h intervals during 16 hours of larval development and processed for bulk RNA-seq and ATAC-seq. This revealed ~1400 genes with oscillations in both accessibility and expression. Remarkably, in most cases the two features were in phase with a ~23 min delay from opening of chromatin to mRNA detection. Comparing the oscillating ATAC peaks with available transcription factor (TF) ChIP-seq data, the authors deduced that the binding of 9 TFs can explain the observed chromatin dynamics. To test the robustness of the developed models, the authors depleted one of these TFs (GRH-1) and found a very good agreement between predicted and measured data.

The manuscript is very well written and convincing. Most methods are described in detail, although certain bioinformatics steps are less clear. For instance, the Materials and Methods section describes the processing of ChIP-seq data from 9 TFs, but the Results section includes a statement about 284 TFs. Ref 25 mentions 217 TFs so what is the source of the remaining TF?

I believe this manuscript will be deeply appreciated by the scientific community, but I suggest to first address the following points:

The authors conclude that seven tissues display pronounced oscillations (marked in red in Fig 1C). However, from visual inspection of the figure it is not obvious that vulval precursors and excretory cells have more circular expression than for instance intestine. Can this be determined in a more objective, numerical manner?

The authors state on page 6 that "The genes that oscillate in both datasets (Fig.1F - red dots) exhibited not only consistent amplitudes but also a remarkable correlation of phases (Fig.1G)." Could this simply be because the single-cell expression profiles were synchronized using the bulk RNA-seq as reference?

Regarding the correlation between amplitudes in ATAC-seq and RNA-seq (Fig 2C), it seems that many more genes did not show this correlation, i.e. genes that have a high amplitude in RNA-seq but not in ATAC-seq and vice versa. How many genes are in each category? Does this affect the concluding remark in the paragraph on page 8?

The test of oscillating ATAC-seq peaks identified in the *daf-6* and *nac-1* loci as regulatory sequences represents a nice validation. The fact that both sequences were able to regulate expression of a minimal promoter with the same timing as the larger promoter fragments argues that their activities are both specific and reflective of the peaks in their native location. Nevertheless, I suggest expanding these experiments:

- 1) How well did the reporters containing the isolated peaks or the promoters reflect the endogenous genes? The plots in Fig 3C and E suggests that this is the case, but the different scales on the x-axis complicates the comparison.
- 2) Did the two peak sequences induce expression in the expected cell types? In other words, do they encode both temporal and spatial information?
- 3) The two other peaks inside the *daf-6* gene have different phases. Are they able to drive expressions in the context of the minimal promoter? In an oscillating manner and if so, with which phase?
- 4) Finally, including a non-oscillating peak upstream of a non-oscillating gene could serve as a relevant negative control.

Reviewer #2:

Summary

This study investigated how the *C. elegans* molting clock schedules stage-specific gene expression. Initial time-series single-cell RNA-seq on synchronized L2-L3 larvae confirmed that broad rhythmic gene expression phase distributions, previously seen in bulk RNA-seq, are inherent to individual oscillating epithelial tissues. This was established using PCA-based pseudotime referenced against bulk RNA-seq, supporting further bulk analysis for mechanistic insights.

Subsequent high-resolution bulk RNA-seq and ATAC-seq on synchronized larvae revealed that rhythmic mRNA transcription

coincides with dynamic, phase-specific changes in chromatin accessibility, functionally validated by reporter assays near the *daf-6* locus. Overlapping oscillating ATAC-seq peaks with TF ChIP-seq data identified 29 enriched TFs. A linear model predicting chromatin rhythmicity from TF binding highlighted nine "molting clock TFs" whose inferred activities, distributed around the cycle, explained most phase/amplitude variance. This model indicated these TFs act largely additively, as interaction terms offered little benefit. The authors concluded that the co-binding of multiple TFs and the additive sum of their vector contributions explain the broad peak-phase dispersion. They also proposed destructive interference between antiphasic TFs as a mechanism for non-rhythmic outputs.

Finally, the model was validated by GRH-1 depletion using an auxin-inducible degron, which disrupted development and affected other TF activities. A timepoint-specific linear model successfully predicted the resultant changes in chromatin accessibility and mRNA expression. The overall conclusion is that these molting clock TFs additively direct the phase of both chromatin accessibility and gene expression.

Overall, this is an important and comprehensive work, supported by a large amount of data and a well-executed series of experiments. The paper is generally well-written, and the core results regarding the additive TF model are convincing. However, there are instances where the methodological approach involves certain shortcuts, particularly concerning some aspects of the scRNA-seq analysis and the subsequent shift in focus away from tissue-specific rhythms after their initial characterization. For example, the significant asynchrony revealed by the scRNA-seq within collection timepoints raises questions about its potential impact on the quantitative interpretation of the bulk sequencing data, which assumes a high degree of population synchrony. Specific parts of the analysis might benefit from the application of more conventional or specialized methods, which could further strengthen the resulting conclusions.

Major points:

1) The title could be improved by specifying the organism and biological context, by highlighting that the study focuses on the *C. elegans* molting clock.

2) Several aspects of the scRNA-seq analysis and interpretation could be clarified or improved:

- For cell type/tissue identification across different time points, it is common to use data integration methods (e.g., Harmony, scVI, Seurat rPCA) to correct for potential batch effects or variations due to experimental conditions, such as the different sampling times used here. Such an approach could help ensure that cells cluster primarily by tissue/cell type rather than by temporal variations, potentially preventing the formation of multiple clusters for the same cell type due to these variations. This might also aid in more robust doublet identification.
- The authors currently use a heuristic approach to identify doublet clusters by assessing if their expression profiles resemble a mix of other clusters. While this has a rationale, more established and cell-centric doublet detection tools are available and could provide a more refined analysis. Applying such methods might lead to cleaner data for cell type identification, reducing confounding effects from time or technical artifacts.
- It would be beneficial to see standard scRNA-seq QC metrics presented, such as visualizations of sample-to-sample heterogeneity, total UMI counts per cell, and the percentage of mitochondrial reads per cell. This would offer a clearer picture of data quality, showing each replicate independently.
- The manuscript does not explicitly show plots of raw or pseudo-bulk scRNA-seq data per cell type as a function of the original sampling time, with distinct replicates visualized (similar to the style of Figure S2D for the pseudo-time approach). Presenting data for some well-known rhythmic genes in this manner could be informative.
- The authors discussed the observed "phase" spread from the time-point specific PCA plots (e.g., Figure S2C) to argue that this is "likely caused by incomplete sample synchrony and/or the lengthy cell isolation procedure", thus necessitating the use of pseudo-time. If the phase spread due to incomplete synchrony is indeed as large as suggested by figures like S2C, this could pose a significant challenge for interpreting the bulk RNA-seq and ATAC-seq data, which inherently average signals from these mixed populations. If the spread is primarily due to variability introduced during sample preparation for the 10x platform, it would be useful to understand if such differences between time points are expected, why it might not manifest as a consistent drift, and why different tissues exhibit varied phase distributions in the PCA plots. Refining the pseudo-time inference method (see below) might lead to more consistent and clearer phase distributions, hopefully more centered on the sampling and homogeneous across tissues.
- Inferring a pseudo-time phase from scRNA-seq data is a known challenge, particularly for cyclical processes like circadian rhythms or the cell cycle, and multiple methods already exist (Tempo, Cyclum, DeepCycle). A common strategy in these methods is to initialize the process using a set of "seed genes" that are well-characterized components of the cycle (e.g., core clock genes or cell-cycle regulators). While the *C. elegans* molting clock might present tissue-specific complexities, the finding that nine TFs can explain much of the rhythmicity suggests a common underlying signal should be detectable. The current method uses the top 2000 highly variable genes, which capture temporal variance (oscillatory and graded) but also other

sources of biological and technical variability. This might explain why many tissues in Figure 1C do not exhibit a clear "circular" structure in the initial PCA plots (though it's possible such structures might appear in other principal component combinations). If the authors wish to continue using PCA to infer pseudo-time and enable comparisons across tissues, they might consider using a consistent set of genes across all tissues, such as established molting clock genes or genes identified as highly rhythmic from their pseudo-bulk analysis.

- The current approach of using bulk RNA-seq data to determine the directionality and set the reference point for the scRNA-seq pseudo-time trajectories appears to contradict the rationale presented in Figure 1A (Scenario 2), which suggests that bulk data can be ambiguous regarding underlying tissue-specific phases.

3) Throughout the manuscript, for the ATAC-seq or RNA-seq, when the authors report rhythmic genes/peaks, phase, and amplitude via harmonic regression ("cosine curve fitting"), it is unclear if they select only statistically significant genes/peaks, for instance, by using a likelihood ratio test versus a null model, and if they correct for multiple testing. The period used for these analyses is also not explicitly stated. For example, in Figure 1E, it is not specified whether the depicted genes are only those determined to be statistically significantly rhythmic. Currently, it is difficult to assess the density of this plot; a density curve or polar histogram would be more informative. If proper thresholds for statistical significance (e.g., FDR) and amplitude were applied, would the phases of the identified rhythmic genes still appear uniformly distributed across the cycle? The same question applies when the authors classify the peaks in three categories (flat, graded and oscillating).

4) In Figure 1D, the authors show that genes tend to oscillate predominantly in one main tissue. Can their linear model, with the nine molting clock TFs, explain this observed tissue specificity? Related to this, could the authors clarify this sentence "oscillating peaks and an additional 753 non-oscillating peaks specific to oscillating tissues". It would be insightful if the authors discussed the expression levels of these nine TFs within the scRNA-seq identified tissues. Furthermore, have the authors considered the potential role of pioneering, tissue-specific factors that might initially open chromatin, thereby enabling the "molting clock TFs" to subsequently drive tissue-specific rhythmic gene expression?

5) Linear regression can be sensitive to outliers, and in this study, the authors use two independent linear regressions for the x and y coordinates. In the original ISMARA model (Balwierz, 2014), a Bayesian approach and cross-validation were employed to avoid overfitting. Similarly, Sobel et al. (reference [26] in the paper) utilized an Elastic-Net penalized linear regression model, fitting x and y components together. Could the authors provide a rationale for their choice of a simpler linear regression approach compared to these more regularized methods?

6) In the GRH-1 depletion experiment, the authors could not measure one full cycle and therefore applied their model in a time-point-specific manner. The cited reference for this method might not be the correct one. It would be useful to clarify if the data was mean-centered before this time-point-specific modeling to ensure the analysis explains sample-to-sample variation rather than the mean expression level of each gene. Additionally, what percentage of variance does this time-point-specific model explain for the GRH-1 depletion data? Would applying this same time-point-specific modeling approach to the control condition data also identify the same nine TFs as the primary drivers?

Minor points:

1. The GEO database indicates three scRNA-seq pools/replicates per timepoint. This and how replicates were handled in analyses (e.g., in pseudo-time inference, curve fitting) should be clearly stated.
2. The sentence: "log₂ transformed and added a pseudocount of 1" would be more precise as: "log₂ transformed after adding a pseudocount of 1."
3. The authors use different pseudo-count values across the dataset, could they justify it?
4. Given that the 10x Genomics protocol was used, the authors should refer to UMI counts rather than read counts.

Reviewer #3:

The authors use single cell RNA sequencing to reveal the gene oscillations mainly stem from a set of seven epithelial tissues. Using ATAC-seq to study oscillating gene expression during development they identify regulatory elements with rhythmically changing chromatin accessibility. Using their data the authors develop a linear model to predict chromatin dynamics based on the binding profiles of approximately two hundred transcription factors. This model allows them to identify nine key regulators

acting additively to determine the peak phase and amplitude of each regulatory element. The model indicates additive action of the transcription factors and suggests a mechanism of non-rhythmic activity through destructive interference. They use their previously established protocol for depletion of GRH-1/Grainyhead to validate some key aspects of their model.

This is a very well-conducted study that provides a wealth of data that should be useful for the field, with interesting predictions deriving from the model. There are key conclusions from the model that would profit from experimental verification as described in the comments below, but if these are not feasible this should not preclude publication.

Major points

1)

The authors devise a model that identifies nine central TFs to be required for the developmental oscillations. Many of these TFs are already known to control rhythmic gene expression. The manuscript would profit from explaining which factors have been shown previously to control oscillations, and which ones haven't. This could be supported by citing respective papers. If there are factors among these nine that have not yet been shown to have phenotypes in oscillation, it would be important to functionally test these factors and show their phenotypes. Using e.g. RNAi in combination with a gene expression reporter readout could be a means to identify phenotypes if this has not yet been demonstrated previously.

2)

An important implication of the model is that there is additivity of factor activity but this is not tested experimentally. Testing this idea experimentally, e.g. via using double mutation and gene expression readouts, could improve the manuscript.

3)

Another important and indeed counterintuitive conclusion from the model is the destructive interference between TFs that can negate oscillations. Again, there is no direct experimental validation for this phenomenon presented. If this could be addressed experimentally e.g. by combining mutation with gene expression assays the manuscript would improve substantially.

Minor points

4)

The authors mention a depletion of neurons and show cholinergic neurons in Fig1. They could elaborate on this in the main text and explain why they use cholinergic neurons specifically for the analysis. Are cholinergic neurons the neuron types that could be extracted most straightforwardly - perhaps due to them constituting the largest group of neurons? Did the authors test subpopulations of cholinergic neurons for oscillations? Are there other neuronal cell types that perhaps show oscillations?

5)

The authors analysed chip seq data for 213 TFs, which is a fraction of the known TFs. It would be useful to state the total number of TFs when describing the analysis, so that it is clear what fraction is analysed.

Reviewer #1:

We thank the reviewer for their positive evaluation of our work and provide below a detailed response to their specific comments.

Gaidatzis and coworkers report a very detailed and carefully executed study on oscillatory gene expression in the nematode *Caenorhabditis elegans*. Rhythmic gene expression is observed in multiple biological systems and play critical roles during development, tissue homeostasis, etc. yet underlying genetic mechanisms are unclear. Here, the authors determined gene expression profiles in individual cells from synchronised *C. elegans* larvae at four different time points. They identified seven cell types that showed oscillatory gene expression. By calculation of pseudo-time gene expression profiles and comparing these with bulk RNA-seq data, the amplitude and phase for each gene were determined. Next, samples were prepared at 1h intervals during 16 hours of larval development and processed for bulk RNA-seq and ATAC-seq. This revealed ~1400 genes with oscillations in both accessibility and expression. Remarkably, in most cases the two features were in phase with a ~23 min delay from opening of chromatin to mRNA detection. Comparing the oscillating ATAC peaks with available transcription factor (TF) ChIP-seq data, the authors deduced that the binding of 9 TFs can explain the observed chromatin dynamics. To test the robustness of the developed models, the authors depleted one of these TFs (GRH-1) and found a very good agreement between predicted and measured data.

The manuscript is very well written and convincing. Most methods are described in detail, although certain bioinformatics steps are less clear. For instance, the Materials and Methods section describes the processing of ChIP-seq data from 9 TFs, but the Results section includes a statement about 284 TFs. Ref 25 mentions 217 TFs so what is the source of the remaining TF?

We downloaded a total of 283 ChIP-seq data sets from the database associated with the manuscript Ref 25 (Kudron, Victorsen et al., 2018). The manuscript was published in March 2018 and the last update to the database was February 2020. We assume that additional datasets were added in the intervening period. Adding our own ChIP-seq dataset for *grh-1* resulted in a total of 284 TFs, which are displayed in Appendix Fig.S6A.

I believe this manuscript will be deeply appreciated by the scientific community, but I suggest to first address the following points:

The authors conclude that seven tissues display pronounced oscillations (marked in red in Fig 1C). However, from visual inspection of the figure it is not obvious that vulval precursors and excretory cells have more circular expression than for instance intestine. Can this be determined in a more objective, numerical manner?

We did not find any strategy to define the set of oscillatory tissues in a fully automated fashion. There were multiple pieces of information that we knew we needed to consider. First, from previous work and from Fig.EV1B, we knew roughly in which tissues to expect oscillations as those tissues expressed genes that oscillate in bulk. Secondly, we needed a circle-like structure in Figure 1C in order to be able to calculate pseudo time. Finally, when synchronizing to the bulk, we wanted to see a clear maximum in Fig.EV1E that would only show in one direction and not the other. We considered this last piece of information critical as it gives us high confidence that the circles in PC space are a true reflection of biological time as opposed to an artifact or a result of a confounder. We now added a sentence to the manuscript for clarification (Materials and Methods: Synchronization of cell-type specific ...). Multiple tissues that showed promising circle-like structures in Figure 1C did not pass that criterion including e.g. intestine (Reviewer Figure 1). In summary, defining our set of oscillating tissues was mainly a result of human curation, weighing multiple pieces of information.

Reviewer Figure 1: Synchronization procedure (similar to Fig.EV1E) applied to intestinal cells, highlighting the absence of a clear maximum correlation in either direction.

The authors state on page 6 that "The genes that oscillate in both datasets (Fig.1F - red dots) exhibited not only consistent amplitudes but also a remarkable correlation of phases (Fig.1G)." Could this simply be because the single-cell expression profiles were synchronized using the bulk RNA-seq as reference?

We can exclude the possibility that the synchronization procedure would create such a high correlation of phases because selecting the optimal starting point and direction is only done once per tissue without affecting the relative timing between genes, as the synchronization synchronizes all the genes within a tissue at the same time. This represents a large constraint. By contrast, the fact that the points scatter along the diagonal (as opposed to an off-diagonal) is a direct consequence of the synchronization procedure.

Regarding the correlation between amplitudes in ATAC-seq and RNA-seq (Fig 2C), it seems that many more genes did not show this correlation, i.e. genes that have a high amplitude in RNA-seq but not in ATAC-seq and vice versa. How many genes are in each category? Does this affect the concluding remark in the paragraph on page 8?

Using the same cutoffs as in Fig 2C (cutoff_RNAseq_amplitude=0.5, cutoff_ATACseq_amplitude=0.25) we obtained 1,118 genes with high amplitude at the RNA-seq level but not at the ATAC-seq level and 1,195 genes with high amplitude at the ATAC-seq level but not at the RNA-seq level. We added these numbers to the manuscript and modified our conclusion on page 8 accordingly.

The test of oscillating ATAC-seq peaks identified in the *daf-6* and *nac-1* loci as regulatory sequences represents a nice validation. The fact that both sequences were able to regulate expression of a minimal promoter with the same timing as the larger promoter fragments argues that their activities are both specific and reflective of the peaks in their native location. Nevertheless, I suggest expanding these experiments:

1) How well did the reporters containing the isolated peaks or the promoters reflect the endogenous genes? The plots in Fig 3C and E suggests that this is the case, but the different scales on the x-axis complicates the comparison.

A comparison of Fig 3C to 3E cannot address this question since they deal with different entities, mRNA vs. fluorescent protein levels, which are expected to differ in stability, and thus in amplitude and peak phase. Hence, to address the reviewer's question, we performed an RT-qPCR-based time course quantifying both endogenous *daf-6* mRNA and the reporter transcript driven by the *daf-6* ATAC-seq element, in the same samples. We find a striking similarity of their dynamics, as shown in the new Appendix Figure S5B.

2) Did the two peak sequences induce expression in the expected cell types? In other words, do they encode both temporal and spatial information?

We added fluorescence images for each reporter construct showing GFP distribution along the animal (Appendix Figure S5A). The ATAC-seq peak constructs show a similar expression pattern as the full promoter constructs, supporting the notion that they encode both temporal and spatial information.

3) The two other peaks inside the *daf-6* gene have different phases. Are they able to drive expressions in the context of the minimal promoter? In an oscillating manner and if so, with which phase?

While an interesting question, we considered the ATAC-seq peaks within promoters as the most likely drivers of gene expression dynamics and thus focused on the two ATAC-seq peaks that we tested. Nevertheless, we are intrigued by oscillating non-promoter ATAC-seq peaks and consider their role in regulating gene expression an important question for a future study.

4) Finally, including a non-oscillating peak upstream of a non-oscillating gene could serve as a relevant negative control.

We did not include such a negative control because we considered having two anti-phase constructs sufficient to assess the ability to drive specific expression dynamics. Non-oscillating peaks would very likely drive expression in non-oscillating tissues, limiting their usefulness as a control for studying oscillations. The minimal promoter itself shows very limited expression and no dynamics (Appendix Figure S5B), suggesting that our ATAC-seq peaks are the main drivers of the expression output.

Reviewer #2:

Summary

This study investigated how the *C. elegans* molting clock schedules stage-specific gene expression. Initial time-series single-cell RNA-seq on synchronized L2-L3 larvae confirmed that broad rhythmic gene expression phase distributions, previously seen in bulk RNA-seq, are inherent to individual oscillating epithelial tissues. This was established using PCA-based pseudotime referenced against bulk RNA-seq, supporting further bulk analysis for mechanistic insights.

Subsequent high-resolution bulk RNA-seq and ATAC-seq on synchronized larvae revealed that rhythmic mRNA transcription coincides with dynamic, phase-specific changes in chromatin accessibility, functionally validated by reporter assays near the *daf-6* locus. Overlapping oscillating ATAC-seq peaks with TF ChIP-seq data identified 29 enriched TFs. A linear model predicting chromatin rhythmicity from TF binding highlighted nine "molting clock TFs" whose inferred activities, distributed around the cycle, explained most phase/amplitude variance. This model indicated these TFs act largely additively, as interaction terms offered little benefit. The authors concluded that the co-binding of multiple TFs and the additive sum of their vector contributions explain the broad peak-phase dispersion. They also proposed destructive interference between antiphasic TFs as a mechanism for non-rhythmic outputs.

Finally, the model was validated by GRH-1 depletion using an auxin-inducible degron, which disrupted development and affected other TF activities. A timepoint-specific linear model successfully predicted the resultant changes in chromatin accessibility and mRNA expression. The overall conclusion is that these molting clock TFs additively direct the phase of both chromatin accessibility and gene expression.

Overall, this is an important and comprehensive work, supported by a large amount of data and a well-executed series of experiments. The paper is generally well-written, and the core results regarding the additive TF model are convincing. However, there are instances where the methodological approach involves certain shortcuts, particularly concerning some aspects of the scRNA-seq analysis and the subsequent shift in focus away from tissue-specific rhythms after their initial characterization. For example, the significant asynchrony revealed by the scRNA-seq within collection timepoints raises questions about its potential impact on the quantitative interpretation of the bulk sequencing data, which assumes a high degree of population synchrony. Specific parts of the analysis might benefit from the application of more conventional or specialized methods, which could further strengthen the resulting conclusions.

We thank the reviewer for their positive evaluation of our work and have followed their suggestions in testing alternative approaches to scRNA-seq analysis, which have largely yielded comparable results, bolstering our claims.

Major points:

1) The title could be improved by specifying the organism and biological context, by highlighting that the study focuses on the *C. elegans* molting clock.

We respectfully disagree with the reviewer on this point. We propose that the more general title, along with the specific information in the abstract, appropriately balance visibility to the diverse audiences interested in *C. elegans* and developmental biology, genetic oscillators, and dynamic gene expression. Nonetheless, we would be happy to defer to the editor on this decision.

2) Several aspects of the scRNA-seq analysis and interpretation could be clarified or improved:

- For cell type/tissue identification across different time points, it is common to use data integration methods (e.g., Harmony, scVI, Seurat rPCA) to correct for potential batch effects or variations due to experimental conditions, such as the different sampling times used here. Such an approach could help ensure that cells cluster primarily by tissue/cell type rather than by temporal variations, potentially preventing the formation of multiple clusters for the same cell type due to these variations. This might also aid in more robust doublet identification.

We apologize for any confusion caused by a lack of clarity on the experimental set-up: the single cell RNA sequencing was performed as a single experiment. This included a total of 4 conditions (4 time points) and in order to obtain sufficient cell numbers at each time point, each of the 4 samples was run in three single 10X channels (Named pools A,B,C on the GEO submission). The pools do not refer to independent experiments. We did not observe systematic differences between the 10X channels for any given sample. Hence, and due to the extensive asynchrony that we observed in every time point, we assumed that batch correcting for time would have little impact on the data. To explore this assertion further, we now performed a test using fastMNN to calculate batch corrected principal components before running UMAP, keeping the rest of the analysis pipeline unmodified. The resulting projection is depicted in Reviewer Figure 2 alongside the original projection, coloring the cells by the previously annotated tissue. We see very little difference between the two projections, especially for the tissues we are mostly interested in. In our opinion this does not warrant the incorporation of a batch correction method, especially given that running UMAP multiple times on the same data will also produce a slightly different result.

Reviewer Figure 2: Comparison of UMAP projections either without (left – original from Fig.1B) or with (right) batch correction.

- The authors currently use a heuristic approach to identify doublet clusters by assessing if their expression profiles resemble a mix of other clusters. While this has a rationale, more established and cell-centric doublet detection tools are available and could provide a more refined analysis. Applying such methods might lead to cleaner data for cell type identification, reducing confounding effects from time or technical artifacts.

We agree that doublet detection can be performed either at the cluster or at the single cell level, and consider both approaches equally well established (see bioconductor tutorial <https://bioconductor.org/books/3.15/OSCA.advanced/doublet-detection.html>). We concluded that the former approach would be appropriate to address the doublet issue in our manuscript, because our experimental setting is somewhat extreme with over 20 tissues in a single sample as a result of profiling a whole organism. The number of doublet tissue combinations is therefore very high ($20 \times 19/2 = 190$). Hence, most tissue combination doublets will not even form clusters with high enough cell counts to pass our cluster size threshold of 100 cells. The only exception would be the muscle tissue because of its high abundance (33% of all the cells). Indeed, the clusters that we identified as doublets involve the muscle cluster 29.

To further test our assumption, we applied a single cell doublet detection method (Dahlin et al. 2018, implemented in the function `computeDoubletDensity()` of the bioconductor package `scDbtFinder` (from the tutorial specified above) and obtained a doublet score for each cell. Reviewer Figure 3 in the top left panel shows the doublet score density for all the cells (black line). The R package also comes with an automatic selection criterion to identify the doublet cells. In our case, the cutoff was set to 0.84 (magenta vertical line in the plot). At this cutoff, there would be 14,222 doublet cells out of 96,609, which seems unreasonably high. Closer inspection of the doublet score distribution (top right panel) revealed a bimodal structure, arguing that a value of at least 7 would be appropriate to isolate the doublet population. To validate this assessment, we also plotted the doublet score density for all the cells that we have previously identified confidently as doublets (clusters 18,23,25,31, red line in top left panel). Here we see a strong shift in the density and a cutoff at approximately 7 (dashed blue vertical line in top left panel) seems also appropriate. We selected doublet cells according to that criterion and highlighted those cells in our UMAP projection, either for the doublets in the manuscript (bottom left panel), or the newly identified doublets by the single cell method (bottom right panel). While we see good agreement between the methods overall, the single cell method identifies more doublets (2,341 vs. 1,212). Many, if not most of the additional doublets are within solid clusters instead of the periphery, raising potential doubts about their real doublet status. Indeed, at the end of the doublet tutorial there is a section urging caution when identifying doublets at the single cell level, arguing that the doublet calls should be interpreted in the light of their respective clusters. Given that only a small number of cells would be affected by changes to our doublet detection methodology and given that we have no ground truth to determine if those changes would lead to an actual improvement, we decided not to change our current doublet detection method.

Reviewer Figure 3: Comparison of cluster-based or cell-based doublet detection approaches. Top, evaluating doublet score thresholds (from scDblFinder) based on cell densities. Bottom, UMAP (from Fig.1B) with doublet cells identified by cluster-based (left) or cell-based (right) highlighted in red.

- It would be beneficial to see standard scRNA-seq QC metrics presented, such as visualizations of sample-to-sample heterogeneity, total UMI counts per cell, and the percentage of mitochondrial reads per cell. This would offer a clearer picture of data quality, showing each replicate independently.

We created Reviewer Figure 4 showing the requested readouts at the single cell level and for each time point (TP1-4) separately. We also added the median UMI count per cell to the main text. We reiterate that we do not have replicate experiments in the scRNA-seq data set.

Reviewer Figure 4: QC metrics for individual time points from the scRNA-seq experiment.

- The manuscript does not explicitly show plots of raw or pseudo-bulk scRNA-seq data per cell type as a function of the original sampling time, with distinct replicates visualized (similar to the style of Figure S2D for the pseudo-time approach). Presenting data for some well-known rhythmic genes in this manner could be informative.

As mentioned above, we do not have replicates in the scRNA-seq data set. In total we have 4 original sampling time points. For a given gene and tissue, we thus could draw 4 gene expression violin plots. Due to the extensive asynchrony present in the scRNA-seq data, these plots would show very little oscillations even for highly oscillating genes.

- The authors discussed the observed "phase" spread from the time-point specific PCA plots (e.g., Figure S2C) to argue that this is "likely caused by incomplete sample synchrony and/or the lengthy cell isolation procedure", thus necessitating the use of pseudo-time. If the phase spread due to incomplete synchrony is indeed as large as suggested by figures like S2C, this could pose a significant challenge for interpreting the bulk RNA-seq and ATAC-seq data, which inherently average signals from these mixed populations. If the spread is primarily due to variability introduced during sample preparation for the 10x platform, it would be useful to understand if such differences between time points are expected, why it might not manifest as a consistent drift, and why different tissues exhibit varied phase distributions in the PCA plots. Refining the pseudo-time inference method (see below) might lead to more consistent and clearer phase distributions, hopefully more centered on the sampling and homogeneous across tissues.

We apologize for the poor choice of words in this section and modified the text. The spread in the scRNA-seq is primarily due to variability introduced during the cell dissociation protocol which is only used to acquire the scRNA-seq data and takes approximately 2-3 hours. We speculate that cells partially continue their development during the dissociation process, leading to non-uniform phase distributions. In contrast, the bulk RNA-seq and ATAC-seq samples are snap frozen immediately after sample collection precluding further development. Extensive asynchrony in bulk samples would have led to massive reduction in gene expression amplitudes in bulk RNA-seq, which we do not observe: Comparing bulk amplitudes to amplitudes from the in-silico reconstituted worm (Figure 1F) shows that the bulk amplitudes are almost as high as the amplitudes from the pseudo-timed scRNA-seq data, arguing that the worms in the bulk experiment are highly synchronous. Furthermore, the high correlation in amplitude as well as

phase gives us confidence that any expression changes that occur during the dissociation protocol still mimic the in-vivo setting. We think that the non-uniform phase distributions are intrinsic to the experimental procedure and not a result of our analytical methodology. This assessment is supported by the presence of non-uniform circular distributions in the initial UMAP projection (Figure 1B), arguing that this is not an artifact of our PCA approach (see below for further discussion).

- Inferring a pseudo-time phase from scRNA-seq data is a known challenge, particularly for cyclical processes like circadian rhythms or the cell cycle, and multiple methods already exist (Tempo, Cyclum, DeepCycle). A common strategy in these methods is to initialize the process using a set of "seed genes" that are well-characterized components of the cycle (e.g., core clock genes or cell-cycle regulators). While the *C. elegans* molting clock might present tissue-specific complexities, the finding that nine TFs can explain much of the rhythmicity suggests a common underlying signal should be detectable. The current method uses the top 2000 highly variable genes, which capture temporal variance (oscillatory and graded) but also other sources of biological and technical variability. This might explain why many tissues in Figure 1C do not exhibit a clear "circular" structure in the initial PCA plots (though it's possible such structures might appear in other principal component combinations). If the authors wish to continue using PCA to infer pseudo-time and enable comparisons across tissues, they might consider using a consistent set of genes across all tissues, such as established molting clock genes or genes identified as highly rhythmic from their pseudo-bulk analysis.

We agree that inferring cyclical trajectories from scRNA-seq data is challenging, especially when there are multiple cyclical processes taking place simultaneously in the multiple tissues. Overall, we pursued the strategy of trying to avoid biases that might be introduced by the custom selection of seed genes. We considered this particularly important in our case, given that before this study, we did not have a confident list of tissues that show rhythmic gene expression, let alone a list of universal seed genes that would show oscillations in all oscillating tissues. Indeed, the lion's share of all the oscillating genes is expressed in a tissue specific fashion (Figure 1D). Nevertheless, using all known oscillating genes in bulk, instead of the top 2000 highly variable genes, to perform the PCA, had very little impact on our circular structures (Reviewer Figure 5, not necessarily the same orientation).

Reviewer Figure 5: PCA scatterplots for oscillating tissues using all known oscillating genes to perform the PCA, instead of the top 2,000 highly variable genes. To be compared with Fig.1C, bearing in mind that the plots are not necessarily in the same orientation.

- The current approach of using bulk RNA-seq data to determine the directionality and set the reference point for the scRNA-seq pseudo-time trajectories appears to contradict the rationale presented in Figure 1A (Scenario 2), which suggests that bulk data can be ambiguous regarding underlying tissue-specific phases.

Indeed, this is true when considering only a single gene, as in the scenario that is depicted in Figure 1A. But in our biological system, there are thousands of oscillating genes, oscillating at different intensity levels, amplitudes and phases. The determination of the directionality and the reference point for the scRNA-seq data is performed once per tissue, not once per gene. This single step needs to synchronize all the genes at the same time, which is a large constraint. Additionally, we are in a fortunate situation in our biological system, where each tissue is characterized by a specific set of oscillating genes (Figure 1D). Selecting the wrong directions or starting points in a given tissue would result in stark discrepancies between the scRNA data and the bulk data, which cannot be compensated for by other tissues due to lack of expression. Since we don't see such discrepancies (Figure 1D), we conclude that the synchronization procedure has worked well for all the tissues.

3) Throughout the manuscript, for the ATAC-seq or RNA-seq, when the authors report rhythmic genes/peaks, phase, and amplitude via harmonic regression ("cosine curve fitting"), it is unclear if they select only statistically significant genes/peaks, for instance, by using a likelihood ratio test versus a null model, and if they correct for multiple testing.

We deliberately chose to select oscillating peaks based on an amplitude cutoff (see "Cosine curve fitting on the ATAC-seq data" in Materials and Methods) instead of using a statistical framework in conjunction with an amplitude cutoff. The reason is as follows: If we use limma to perform a cosine fit in the ATAC-seq data (including multiple testing correction), we obtain 8,982 oscillating peaks at an FDR of 5%. A large fraction of those have very small amplitudes (as low as 0.095). This is a negative side effect of having so much such data, namely $17 \times 2 = 34$ data points for only 3 coefficients (sin, cos, gradedCoeff) plus intercept, leading to excessive statistical power. We do not put a lot of trust in those small amplitude peaks and as a result we need to also filter for effect size (amplitude). To select a reasonable amplitude cutoff, we performed cosine fits for the two replicates separately and plotted the resulting amplitudes in a scatter plot. Unfortunately, this plot was not part of the manuscript, but we included it now as Figure EV2E. From visual inspection of this plot, we decided that an amplitude cutoff of 0.25 (in the combined replicate fit using all the data) is the lowest we can reasonably use. At that level, we obtained a total of 5,193 oscillating ATAC-seq peaks, out of which virtually all (5,160) passed the 5% FDR threshold from above. This means that the amplitude cutoff almost completely overrides the FDR calculation, rendering the latter useless. We thus decided to drop that filter altogether. The excessive data argument also holds true in the case of the single cell RNA-seq as well as the bulk RNA-seq data. We therefore always selected oscillating peaks/genes by amplitude only, resulting in much more conservative lists as compared to using a statistical framework.

The period used for these analyses is also not explicitly stated.

We did not specify a period, because the cosine fits were performed in pseudo time space (see "Cosine curve fitting on the ATAC-seq data" in Materials and Methods). This allowed us to perform one single cosine fit for both replicates, even though they are not in perfect synchrony. For this we evaluated the cosine and sine basis functions at the respective pseudo time points in radians.

For example, in Figure 1E, it is not specified whether the depicted genes are only those determined to be statistically significantly rhythmic.

In Figure 1E all genes are depicted, non-oscillating genes are located in the center of the respective plot.

Currently, it is difficult to assess the density of this plot; a density curve or polar histogram would be more informative. If proper thresholds for statistical significance (e.g., FDR) and amplitude were applied, would the phases of the identified rhythmic genes still appear uniformly distributed across the cycle?

While we see broad phase dispersion in Figure 1E in every tissues, the distribution does not seem to be uniform. For example, vulva precursors show an enrichment at roughly 135 degrees. The other tissues also also show some inhomogeneity, even dependent on amplitude. For example, in excretory cells, high amplitude ($\log_2 \text{ampl} > 5$) genes are enriched at 135 degrees whereas lower amplitude genes ($2 < a < 4$) are enriched at 315 degrees. By plotting the data as raw radar plots, we can visualize both, the broad phase dispersion in each tissue as well as the amplitude associated complexity. We consider this an interesting structure in the data that we didn't want to hide, but we also did not want to discuss this in the manuscript as we feel this would be an unnecessary distraction in the already complex single cell analysis section.

The same question applies when the authors classify the peaks in three categories (flat, graded and oscillating).

Also in Figure 2B we see a similar type of inhomogeneity, dependent on the amplitude. We therefore prefer to keep the raw radar plots.

4) In Figure 1D, the authors show that genes tend to oscillate predominantly in one main tissue. Can their linear model, with the nine molting clock TFs, explain this observed tissue specificity?

The model would not be able to recapitulate the tissue specificity because it was only trained to predict amplitude and phase, not average intensity. The different tissue identities are defined by tissue specific transcription factors, most of which are not included in our 9 molting clock TFs. We do not see how the model would be able to predict differences in the average intensity across tissues without incorporating information about a large number of tissue-specific TFs. Nevertheless, we find the idea by the reviewer to apply the TF model to individual tissues very exciting and already started a project to explore this further in a potential follow-up study.

Related to this, could the authors clarify this sentence "oscillating peaks and an additional 753 non-oscillating peaks specific to oscillating tissues".

These can be considered control peaks, peaks that do not oscillate in oscillating tissues. We define those as peaks that do not oscillate at the bulk ATAC-seq level but at the same time are accessible only in oscillating tissues. If they were accessible in non-oscillating tissues, but also oscillating in oscillating tissues, they would accumulate "contaminating" reads from the non-oscillating tissues, leading to amplitude damping at the bulk ATAC-seq level. These peaks would not serve as proper non-oscillating controls because they could still oscillate in oscillating tissues even though we don't observe oscillations at the bulk level. To circumvent that issue, we used data from a publicly available scATAC-seq experiment to ensure that our control peaks are only accessible in oscillating tissues.

It would be insightful if the authors discussed the expression levels of these nine TFs within the scRNA-seq identified tissues.

We included a figure showing the expression of the nine TFs in the sc-RNA-seq tissues (Appendix Figure S6B). This shows that those factors are generally expressed in the oscillating tissues, in particular the ones that are well covered in the ATAC-seq experiment (Pharynx excluded for technical reasons as discussed, see Appendix Figure S4).

Furthermore, have the authors considered the potential role of pioneering, tissue-specific factors that might initially open chromatin, thereby enabling the "molting clock TFs" to subsequently drive tissue-specific rhythmic gene expression?

We consider this an interesting question, but this would require a transcription factor model that predicts amplitude, phase and average expression across different tissues (as also discussed at the beginning of question 4). This is more challenging but we are in the process of exploring this possibility for a follow up study.

5) Linear regression can be sensitive to outliers, and in this study, the authors use two independent linear regressions for the x and y coordinates. In the original ISMARA model (Balwierz, 2014), a Bayesian approach and cross-validation were employed to avoid overfitting. Similarly, Sobel et al. (reference [26] in the paper) utilized an Elastic-Net penalized linear regression model, fitting x and y components together. Could the authors provide a rationale for their choice of a simpler linear regression approach compared to these more regularized methods?

Regularization schemes create additional constraints on the coefficients, L1 regularization is typically introduced to force a model to select only a small number of important features. In such a setting, it would not be surprising for a model to report a small number of important features, it would do so by design. In our case though, we don't introduce such additional constraints and the model still predicts only a small number of relevant factors. This allows us to report this result as an actual finding. We would not be able to do so if we added a regularization scheme. In addition, as the reviewer points out, regularization complicates the situation because it couples the coefficients of the two models for x and y. Therefore, the two models cannot be fit independently. About a year ago, we were in contact with Felix Naef, senior author of the Sobel et al. paper, and he specifically urged us to use caution in the case we would like to include regularization ("Are you planning to use L1 regularization? if so it's a bit more tricky"). In summary we did not see any benefit in including regularization. L1 regularization would have prevented us from making an important finding in the manuscript, L2 regularization would not have been helpful in avoiding overfitting because we have more than enough data to eliminate that concern (3,935 data points for 213 coefficients) while at the same time regularization would have potentially exposed us to a methodological risk. Nevertheless, to rule out the possibility of overfitting, we randomly split the data into two groups and recalculated the p-values from

Figure 4C in both groups separately. Reviewer Figure 6 depicts a scatter plot comparing the two $-\log_{10}$ p-values, showing a very strong correlation.

Reviewer Figure 6: Scatterplot comparing the two p-values obtained for each of the 213 TFs, obtained from two independent linear models (similar to Fig.4C), performed after randomly splitting the data into two groups.

6) In the GRH-1 depletion experiment, the authors could not measure one full cycle and therefore applied their model in a time-point-specific manner. The cited reference for this method might not be the correct one.

We checked again but from what we can tell, in the Rey et al. paper (PMID: 21364973) the authors used same model as we did, applied to individual time points in the circadian system. The formula used is stated in the section "Inference of Transcription Factor Activities". A_{mt} denotes the inferred TF activity matrix with m referring to the motif and t referring to time. So for each motif, they have one activity value per time point, which is the result of fitting one model per time point.

It would be useful to clarify if the data was mean-centered before this time-point-specific modeling to ensure the analysis explains sample-to-sample variation rather than the mean expression level of each gene.

Yes, the data was mean-centered with respect to one cycle in the control condition. We clarified this now in the text.

Additionally, what percentage of variance does this time-point-specific model explain for the GRH-1 depletion data?

We assume the reviewer here refers to the explained variance in the individual conditions (before calculating the deltas that are depicted in Appendix Figure S7D), at the single ATAC-seq peak level. The model explains 37.1% of the variance in the vehicle condition and 33.1% of the variance in the auxin condition when considering all the time points.

Would applying this same time-point-specific modeling approach to the control condition data also identify the same nine TFs as the primary drivers?

If we understood correctly, the question here is whether an individual time point model would have selected the same factors as our x/y model, on wild-type animals. To address this, we sought to compare the two modeling approaches using the same data, namely the original wild-type time course, instead of using the GRH-1 control experiment. Determining a combined significance for every TF after fitting one model per time point poses a methodological challenge, in part because neighboring time points as well as time points that are one period apart are highly correlated. Combining the p-values of the individual time points is not as simple as combining the two orthogonal p-values from our two x and y models. Nevertheless, an ad hoc solution would be to simply select for every TF the lowest p-value at any time point. This would of course be statistically problematic as this would introduce p-value inflation due to multiple testing, but the relative importance of the factors would likely remain

intact. Reviewer Figure 7 compares the p-values from Figure 4C to the ones we obtained here from the individual time point models. Overall, there is very high agreement between the two methods, especially for the most significant TFs. As we go down the list, differences start to emerge, as expected since the minimum p-value selection is extremely sensitive to outliers. Given this result we are confident that our x/y model is an appropriate choice.

Reviewer Figure 7: Scatterplot comparing the two p-values obtained for each of the 213 TFs, obtained from two independent linear models. P-values on the x-axis are from Fig.4C. P-values on the y-axis are from a time-point-specific model, taking the lowest p-value at any time point for each TF.

Minor points:

1. The GEO database indicates three scRNA-seq pools/replicates per timepoint. This and how replicates were handled in analyses (e.g., in pseudo-time inference, curve fitting) should be clearly stated.

The pools A,B,C represent different 10X runs for a given sample. We clarified the description of the GEO submission

2. The sentence: "log2 transformed and added a pseudocount of 1" would be more precise as: "log2 transformed after adding a pseudocount of 1".

We thank the reviewer for noticing that mistake and corrected the respective sentence

3. The authors use different pseudo-count values across the dataset, could they justify it?

We typically normalized to average library size and added a pseudo count of 8. We did that for the RNA-seq as well as the wild-type ATAC-seq. In the GRH-1 ATAC-seq experiment, we needed to add a higher pseudo count of 32 to reduce the variability of low intensity peaks: In the wild-type ATAC-seq experiment, we performed cosine fits considering all the time points. This procedure was able to deal with a substantial amount of noise. However, in the GRH-1 experiment we needed to evaluate the model performance at the level of single ATAC-seq peaks at single time points, which required additional noise reduction for low intensity ATAC-seq peaks by adding a higher pseudo count. In the scRNA-seq experiment, after having inferred tissue- and time dependent expression profiles, we normalized to a total of 1 million reads per sample and time point given that there were, on average, 0.5M reads per oscillating tissue and time point. Here we used a rather lenient pseudo count of 1 to avoid potential damping of oscillations.

4. Given that the 10x Genomics protocol was used, the authors should refer to UMI counts rather than read counts.

Thank you for the suggestion; we changed the text accordingly.

Reviewer #3:

The authors use single cell RNA sequencing to reveal the gene oscillations mainly stem from a set of seven epithelial tissues. Using ATAC-seq to study oscillating gene expression during development they identify regulatory elements with rhythmically changing chromatin accessibility. Using their data the authors develop a linear model to predict chromatin dynamics based on the binding profiles of approximately two hundred transcription factors. This model allows them to identify nine key regulators acting additively to determine the peak phase and amplitude of each regulatory element. The model indicates additive action of the transcription factors and suggests a mechanism of non-rhythmic activity through destructive interference. They use their previously established protocol for depletion of GRH-1/Grainyhead to validate some key aspects of their model.

This is a very well-conducted study that provides a wealth of data that should be useful for the field, with interesting predictions deriving from the model. There are key conclusions from the model that would profit from experimental verification as described in the comments below, but if these are not feasible this should not preclude publication.

We thank the reviewer for the positive evaluation of our work and endorsement of its publication. We particularly appreciate their suggestions for further experimental tests. Unfortunately, as somewhat anticipated by the reviewer, these experiments are not feasible with our current experimental set-up, but we hope to be able to develop suitable alternatives in the future.

Major points

1)

The authors devise a model that identifies nine central TFs to be required for the developmental oscillations. Many of these TFs are already known to control rhythmic gene expression. The manuscript would profit from explaining which factors have been shown previously to control oscillations, and which ones haven't. This could be supported by citing respective papers.

The reviewer is absolutely correct, several of the TFs have been implicated in the control of rhythmic gene expression, mostly through their involvement in controlling rhythmic molting, and we now explicitly mention that this is true for BLMP-1, NHR-23, GRH-1, and NHR-25, and not for the other five factors. We would like to point out, however, that even for the former four factors, our understanding on how and to what extent they affect rhythmic gene expression is currently still quite limited.

If there are factors among these nine that have not yet been shown to have phenotypes in oscillation, it would be important to functionally test these factors and show their phenotypes. Using e.g. RNAi in combination with a gene expression reporter readout could be a means to identify phenotypes if this has not yet been demonstrated previously.

We agree that this is an interesting question but consider it outside the scope of the current paper: Not all reporters would be bound by every TF, requiring many reporters to cover all the 9 TFs. To be convincing, one TF would need to affect multiple reporters, further increasing the number of reporters that would need to be tested. We cannot currently perform the time-lapse imaging experiments at the required scale and throughput. For the future, we

consider acute depletion coupled with time-resolved genomics (ATAC-seq or RNA-seq), like we have done for *grh-1*, a more promising route.

2)

An important implication of the model is that there is additivity of factor activity but this is not tested experimentally. Testing this idea experimentally, e.g. via using double mutation and gene expression readouts, could improve the manuscript.

We assume this refers to experiments performed with reporter constructs that contain specific TF binding sites that are mutated in different combinations. Unfortunately, this would require a large number of reporters because our motif-only model (Figure 4F) has limited predictive power to be reliable at the single gene level. The only way we can overcome this uncertainty is to scale up our reporter system. We agree this would be an important point and we are considering methodological development to address this issue in the future.

3)

Another important and indeed counterintuitive conclusion from the model is the destructive interference between TFs that can negate oscillations. Again, there is no direct experimental validation for this phenomenon presented. If this could be addressed experimentally e.g. by combining mutation with gene expression assays the manuscript would improve substantially.

We agree that direct testing of this model through perturbational or synthetic biology approaches would be valuable. Currently, it is precluded by the technical challenges discussed for point 2 above, but we will seek to develop suitable approaches in the future.

Minor points

4)

The authors mention a depletion of neurons and show cholinergic neurons in Fig1. They could elaborate on this in the main text and explain why they use cholinergic neurons specifically for the analysis.

In our scRNA data, we only found a single cluster that corresponds to neurons, namely cluster 14 containing a total of 1,231 cells (Figure EV1A). The assignment of this cluster to the subtype of “cholinergic neurons” is based on the maximum correlation.

Are cholinergic neurons the neuron types that could be extracted most straightforwardly - perhaps due to them constituting the largest group of neurons? Did the authors test subpopulations of cholinergic neurons for oscillations? Are there other neuronal cell types that perhaps show oscillations?

Indeed, cholinergic neurons are among the most abundant neuronal subtypes ([10.1126/science.aam8940](https://doi.org/10.1126/science.aam8940)) in *C. elegans*, which is probably the main reason why we detected them in our experiment. In our UMAP projection (Figure 1B), we did not see subpopulations of neurons, but we cannot exclude the possibility that another clustering approach would be able to dissect the neuronal population even further. Overall, we believe, that neurons are unlikely to express a large fraction of oscillating genes, because neuron-specific genes are depleted for oscillating genes (<https://doi.org/10.15252/msb.20209498>). We emphasize that this does not preclude the possibility that some genes will be rhythmically expressed in neurons and/or contribute to shaping overall oscillations in larvae.

5)

The authors analysed chip seq data for 213 TFs, which is a fraction of the known TFs. It would be useful to state the total number of TFs when describing the analysis, so that it is clear what fraction is analysed.

We added an estimate of the total number of TFs in *C. elegans* (763) in the text. We used ChIP-seq data from a total of 284 TFs (Appendix Figure S6A), out of which 213 satisfied our criterion to overlap at least 10 oscillating ATAC-seq peaks, qualifying them for inclusion in the modeling.

9th Sep 2025

Manuscript Number: MSB-2025-13069R
Title: A scheduler for rhythmic gene expression
Author: Dimos Gaidatzis
Maike Graf-Landua
Stephen Methot
Michaela Wölk
Giovanna Brancati
Yannick Hauser
Milou Meeuse
Smita Nahar
Kathrin Braun
Marit van der Does
Sirisha Aluri
Hubertus Kohler
Sebastien Smallwood
Helge Großhans

Dear Helge,

Thank you for sending us your revised manuscript. We have now heard back from the two reviewers who were asked to re-evaluate your study. As you will see, the reviewers are satisfied with the modifications made and think that the study is now suitable for publication. Before we can formally accept your manuscript, we would ask you to address the following editorial-level issues:

1. Please provide up to five key words in the manuscript file.
2. Please remove the "Authors' contributions" section from the manuscript file.
3. Funding information: Please verify that the funding details (including project numbers) in the manuscript match those entered in the submission system. Please note that any funding information included in the Comments box cannot be extracted by our production team. Therefore, all funders must be added to the "More Funders" list in the system.
4. Appendix:
 - Please add the missing page number for Appendix Table S1 in the Table of Contents.
 - The table should be renamed to "Appendix Table S1" in both the table legend and the in-text callout.
 - There is a file named "Appendix Tables S1-S3." Please double-check its contents. I feel that Appendix Tables S1-S3 can be removed. Appendix Table S4 should be moved to the Appendix file and renamed "Appendix Table S2." The corresponding in-text callouts should be updated accordingly. This file should then be deleted.
5. The code availability information should be incorporated into the "Data Availability" section, and the separate heading "Code Availability" should be removed.
6. "Competing interests" should be renamed to "DISCLOSURE AND COMPETING INTERESTS STATEMENT".
7. Please ensure all callouts are listed sequentially. Add missing callout for Appendix Table S1.
8. Sections need to be named and the order should be corrected as the following: Title page - Abstract - Keywords - Introduction - Results - Discussion - Methods - Data Availability - Acknowledgements - Disclosure and Competing Interests Statement - References - Figure Legends - Table(s) - Expanded View Figure Legends.

When you resubmit your manuscript, please download our CHECKLIST (<https://bit.ly/EMBOPressAuthorChecklist>) and include the completed form in your submission. *Please note* that the Author Checklist will be published alongside the paper as part of the transparent process (<https://www.embopress.org/page/journal/17444292/authorguide#transparentprocess>)

Click on the link below to submit your revised paper.

Thank you for submitting this interesting paper to Molecular Systems Biology.

Kind regards,
Jingyi

Jingyi Hou, PhD
Senior Editor
Molecular Systems Biology

*** PLEASE NOTE *** As part of the EMBO Press transparent editorial process initiative (see our Editorial at <https://dx.doi.org/10.1038/msb.2010.72> , Molecular Systems Biology will publish online a Review Process File to accompany accepted manuscripts. When preparing your letter of response, please be aware that in the event of acceptance, your cover letter/point-by-point document will be included as part of this File, which will be available to the scientific community. More information about this initiative is available in our Instructions to Authors. If you have any questions about this initiative, please contact the editorial office (msb@embo.org).

Reviewer #1:

I congratulate the authors on their revised manuscript: they have fully addressed my questions and have added new data that strengthen the conclusions.

Reviewer #2:

The authors have adequately addressed our concerns.

All editorial and formatting issues were resolved by the authors.

19th Sep 2025

Manuscript number: MSB-2025-13069RR

Title: A scheduler for rhythmic gene expression

Dear Helge,

Thank you again for sending us your revised manuscript. We are now satisfied with the modifications made and I am pleased to inform you that your paper has been accepted for publication.

Kind regards,
Jingyi

Jingyi Hou, PhD
Senior Editor
Molecular Systems Biology
